# Proteostasis perturbation of N-Myc leveraging HSP70 mediated protein turnover improves treatment of neuroendocrine prostate cancer

Pengfei Xu[1,9], Joy C. Yang [1,9], Bo Chen[1,2,9], Shu Ning[1], Xiong Zhang[3], Leyi Wang[1,4], Christopher Nip[1], Yuqiu Shen[1], Oleta T. Johnson [5], Gabriela Grigorean [6], Brett Phinney [6], Liangren Liu[2], Qiang Wei[2], Eva Corey [7], Clifford G. Tepper [3,8], Hong-Wu Chen[3,8], Christopher P. Evans[1,8], Marc A. Dall'Era[1,8], Allen C. Gao[1,8], Jason E. Gestwicki [5] & Chengfei Liu [1,4,8] ✉

N-Myc is a key driver of neuroblastoma and neuroendocrine prostate cancer (NEPC). One potential way to circumvent the challenge of undruggable N-Myc is to target the protein homeostasis (proteostasis) system that maintains N-Myc levels. Here, we identify heat shock protein 70 (HSP70) as a top partner of N-Myc, which binds a conserved "SELILKR" motif and prevents the access of E3 ubiquitin ligase, STIP1 homology and U-box containing protein 1 (STUB1), possibly through steric hindrance. When HSP70's dwell time on N-Myc is increased by treatment with the HSP70 allosteric inhibitor, STUB1 is in close proximity with N-Myc and becomes functional to promote N-Myc ubiquitination on the K416 and K419 sites and forms polyubiquitination chains linked by the K11 and K63 sites. Notably, HSP70 inhibition significantly suppressed NEPC tumor growth, increased the efficacy of aurora kinase A (AURKA) inhibitors, and limited the expression of neuroendocrine-related pathways.

Imbalances in protein homeostasis (Proteostasis) are associated with a wide range of diseases, including neurodegeneration and tumorigenesis[1–4]. In tumors, dysregulated proteostasis leads to aberrant protection of oncoproteins, allowing cancer cells to escape from apoptosis and promoting tumor growth, metastasis, and drug resistance[3,5]. Therefore, one important goal is to identify the key proteostasis factors and find strategies to correct proteostasis imbalance.

Prostate cancer has proven to be a key model in which to understand how proteostasis regulates tumor progression. Castration-resistant prostate cancer (CRPC) can transition to a lineage with reduced androgen receptor (AR) expression which is characterized by up-regulated neuroendocrine markers and characteristics of neuronal identity. The resulting disease, neuroendocrine prostate cancer (NEPC), is aggressive and it has poor prognosis. Specifically, treatment-induced NEPC (t-NEPC) accounts for 20% of CRPC and is characterized by rapid proliferation and metastasis with no treatment options[6–16]. The amplification of N-Myc (encoded by *MYCN*) is a key modification of NEPC[17,18]. N-Myc is capable of suppressing AR-signaling and driving lineage plasticity, tumor aggressiveness, and AR-independent progression in prostate cancer preclinical models[18–20]. However, it has

[1]Department of Urologic Surgery, University of California, Davis, CA, USA. [2]Department of Urology, West China Hospital, Sichuan University, Sichuan, China. [3]Department of Biochemistry and Molecular Medicine, School of Medicine, University of California, Davis, CA, USA. [4]Graduate Group in Integrative Pathobiology, University of California, Davis, CA, USA. [5]Department of Pharmaceutical Chemistry, University of California, San Francisco, CA, USA. [6]Proteomics Core Facility, University of California, Davis, CA, USA. [7]Department of Urology, University of Washington, Washington, WA, USA. [8]University of California, Davis Comprehensive Cancer Center, Sacramento, CA, USA. [9]These authors contributed equally: Pengfei Xu, Joy C. Yang, Bo Chen. ✉e-mail: cffliu@ucdavis.edu

proven challenging to identify therapeutic strategies that reduce N-Myc transcriptional activity or levels.

Here, we sought to take a different approach and identify the molecular chaperones that might be involved in N-Myc proteostasis. Utilizing label-free mass spectrometry, we identified HSP70, specifically the stress-induced form, HSPA1A/HSPA1B, as among the predominant chaperone proteins binding to N-Myc. Further investigation revealed that HSP70 collaborates with STUB1, a co-chaperone protein, and E3 ubiquitin ligase, to mediate the ubiquitination of N-Myc protein. Consistent with this mechanism, the allosteric inhibition of HSP70 by JG231 facilitated the STUB1-dependent ubiquitination and subsequent turnover of N-Myc. Our in vivo animal studies provided validation for the inhibitory effect of JG231 in limiting the growth of NEPC tumors and enhancing the efficacy of the aurora kinase A inhibitor, alisertib. These findings underscore the importance of the HSP70/STUB1 complex in mediating N-Myc protein turnover and highlight its potential as a promising therapeutic strategy for the treatment of NEPC.

## Results

### Patient-derived prostate xenograft tumor after repeated castration-relapse passages exhibits neural lineage plasticity and high levels of N-Myc protein

t-NEPC is mainly caused by lineage plasticity transformation of adenocarcinoma after long-term androgen deprivation therapy or AR signaling inhibitor (ARSI) treatment, such as enzalutamide. We established multiple conditional-reprogrammed cell (CRC) and patient-derived xenograft (PDX) models from patients with high Gleason scores and/or at the castration-resistant stages[21]. Among these models, one spontaneously indefinite cell line, UCDCaP (from a Gleason 10 patient), was developed. We then established the UCDCaP-CR line by passing it through two castration-regression-relapse cycles and three castrated-mouse passages (Fig. 1a and Supplementary Fig. 1a). Immunohistochemical staining revealed strong AR staining in both the patient and PDX specimens (Fig. 1b). Genomic DNA extracted from the patient sample, mouse PDX tumors, and UCDCaP cell cultures were subjected to whole-exome sequencing (WES) to define the repertoire of somatic mutations. A total of 1,062 mutations present in COSMIC (Catalog of Somatic Mutations in Cancer) genes were identified in the patient (n = 779), PDX (n = 953), and/or the UCDCaP cell line (n = 849). The relationships of the individual mutation profiles were visualized using a Venn diagram (Fig. 1c) and demonstrated significant conservation of the mutations present in the original patient sample (n = 736/779; 94.5%) in both the PDX and the UCDCaP cell line. Additionally, the PDX and the UCDCaP cell line contained 191 and 82 mutations that were unique to each model, respectively. Functional enrichment analysis of the common set of mutated genes demonstrated overrepresentation of various pathways, including apoptosis, cell cycle, androgen receptor, EGFR signaling, PI3K/AKT signaling, HIPPO signaling, P53 downstream pathway, and DNA damage response. Accordingly, the patient, PDX, and UCDCaP cells contained mutations in critical genes such as *AR* (missense, deletion), *ATM*, *BRCA1*, *CHD4*, *ERBB2*, *PIK3CA*, *TMPRSS2*, *TP53*, *TSC1/2*, and *XPA* (frameshift) (Supplementary Fig. 1b), again underscoring the relevance and fidelity of the UCDCaP cell line model in preserving the genetic landscape of the original patient tumor. To further characterize the UCDCaP and UCDCaP-CR cells, we performed RNA-seq using cells cultured in charcoal-stripped fetal bovine serum (CS-FBS) conditions. Compared to UCDCaP cells, neural lineage and stem cell differentiation genes were significantly altered in UCDCaP-CR cells (Fig. 1d). Gene Ontology (GO) analysis showed that multiple pathways related to neuronal development and neurotransmitters were significantly upregulated in the UCDCaP-CR cells. Consistently, Kyoto Encyclopedia of Genes and Genomes (KEGG) and Reactome analyses showed that the neuroactive ligand-receptor and NCAM1 interaction pathways

were significantly upregulated in UCDCaP-CR cells (Fig. 1e). Furthermore, Gene Set Enrichment Analysis (GSEA) revealed that neural stem cell differentiation and lineage-specific pathways, signaling pathways regulating pluripotency of stem cells, regulation of neuron projection development, neural lineage-associated genes, and Rickman's 966 N-Myc bivalent genes[17] were highly enriched in UCDCaP-CR cells (Fig. 1f). In addition, the genes from neuropeptide receptor binding, neurotransmitter receptor activity, axon initial segment and neuroactive ligand receptor interaction pathways were significantly enriched in the UCDCaP-CR cells. We also found a significant downregulation of PTEN gene transcription, AR response, P53 signaling, and apoptosis pathways in UCDCaP-CR cells (Supplementary Fig. 1c). Neuroendocrine (NE) genes, such as CHGA, SYP, ENO2, and NKX2-1, were significantly upregulated in UCDCaP-CR cells (Fig. 1g and Supplementary Fig. 1d). qRT-PCR and western blotting confirmed that the NE feature genes were upregulated in UCDCaP-CR cells (Fig. 1h). Surprisingly, mRNA expression of MYCN only increased 1.5 folds, but the protein level of N-Myc was drastically increased in UCDCaP-CR cells, accompanied by NSE and SYP protein overexpression. However, the AR and c-Myc expression levels were decreased.

We then screened multiple prostate cancer cell lines and PDX tumors to determine N-Myc protein expression. These results showed that CWR22Rv1, PC3, H660 cells expressed high levels of N-Myc and NE markers. In addition, PC3 and H660 cells expressed very low level of c-Myc (Fig. 1i). None of the CRPC PDX tumors expressed N-Myc but they had high level of c-Myc. Three out of four NEPC PDX tumors (LuCaP49, LuCaP93, LuCaP145.2, and LuCaP173.1) expressed high level of N-Myc and c-Myc, while LuCaP145.2 only expressed high level of L-Myc, suggesting the possible complementary role of Myc family in CRPC and NEPC transition (Fig. 1j). Immunohistochemistry (IHC) staining of the sections from tumors UCDCaP and UCDCaP-CR, compared to those from LuCaP35CR and LuCaP93, further confirmed the elevated expression of N-Myc in LuCaP93 and UCDCaP-CR tumors (Fig. 1k). RNA-seq data analysis from the collections of prostate cancer patient samples[11] validated the upregulation of the MYCN gene in the NEPC cohort, but not in AR-high prostate cancer (ARPC), AR-low prostate cancer (ARLPC), amphicrine prostate cancer (AMPC), and double negative prostate cancer (DNPC). However, in the PDX samples[22,23], MYCN was upregulated in both DNPC and NEPC cohorts, but not in castration-sensitive prostate cancer (CSPC), CRPC, and AMPC (Fig. 1l). Consistently, the expression of AR and KLK3 was increased in the ARPC, CSPC, CRPC, and AMPC cohorts, whereas SYP and CHGA were highly expressed in AMPC and NEPC (Supplementary Fig. 1e). Therefore, these results demonstrate that long-term androgen deprivation in prostate xenograft tumors exhibits neural lineage plasticity and high levels of N-Myc protein.

### HSP70/STUB1 binds N-Myc protein and controls its levels through the ubiquitin-proteasome system

We next asked which of the NEPC cell lines relied on N-Myc for cell growth. This question was important because we wanted to study chaperone binding to N-Myc in cells whose viability was sensitive to N-Myc proteostasis. Through transfection with N-Myc siRNA, we found that a significant reduction in cell growth was observed in H660, UCDCaP-CR, and CWR22Rv1 cells. We confirmed that the knockdown of N-Myc did not affect the expression of other genes, such as c-Myc and AURKA (Fig. 2a and Supplementary Fig. 2a, b). To understand whether N-Myc was degraded by the ubiquitin-proteasome system (UPS) in these models, we treated the cells with MG132 and found that the N-Myc level was significantly increased in H660 and CWR22Rv1 cells (Fig. 2b). Thus, these cells emerged as suitable models in which growth relied on N-Myc and the protein was turned over through the UPS.

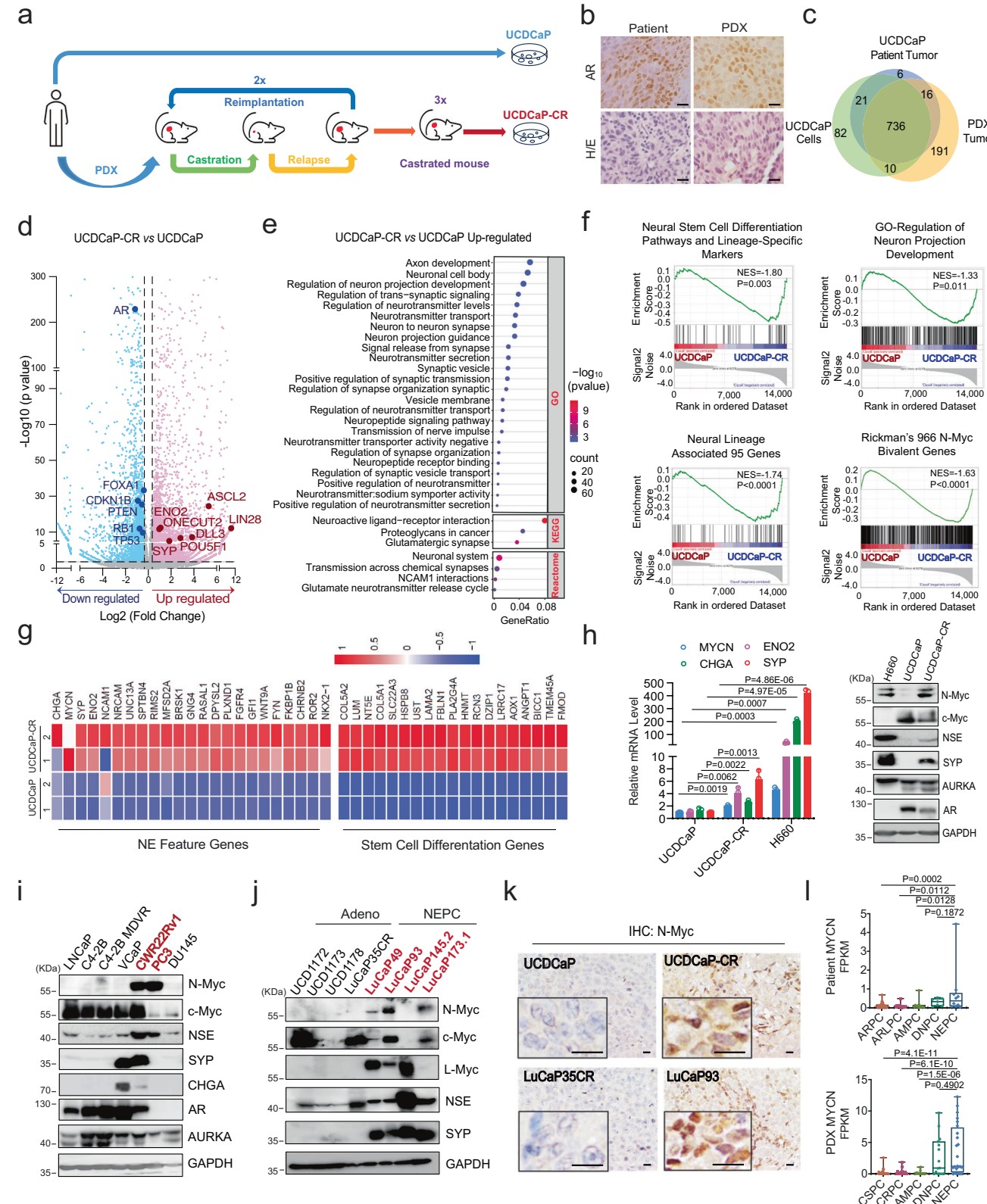

There are ~180 proteins associated with proteostasis[24]. To identify the subset of factors that bind N-Myc, we performed immunoprecipitation pull-down assays, followed by label-free Mass Spectrometry profiling. A total of 779 proteins were identified as N-Myc-binding proteins (protein quantification > 100 compared to the negative control group). Among them, several HSP70 family proteins, such as HSPA1B (HSP70), HSPA6, and HSPA8

(HSC70), were the most enriched (Fig. 2c, d). The HSP70/STUB1 interaction has been recognized as a promising target for therapeutic intervention[25]. Co-immunoprecipitation (Co-IP) experiments revealed that N-Myc was readily detectable in pull-down complexes of exogenous HSP70 and STUB1 in HEK293 cells (Fig. 2e, f). Furthermore, endogenous interaction between N-Myc and HSP70 was confirmed in CWR22Rv1 and UCDCaP-CR cells

**Fig. 1 | Patient-derived prostate xenograft tumor after repeated castration-relapse passages exhibits neural lineage plasticity and overexpresses N-Myc.** **a** Flow chart showing the establishment of the UCDCaP and UCDCaP-CR cell lines. **b** Androgen receptor (AR) expression in both patient and patient-derived xenograft (PDX) specimens by immunohistochemical staining. Scale bar represents 20 microns. **c** Venn diagram showing the mutations in the patient sample, mouse PDX tumor, and UCDCaP cells by whole exome sequencing. **d** Volcano plot showing up-regulated (red) and down-regulated (blue) genes in UCDCaP-CR cells compared with UCDCaP cells. **e** Gene Ontology (GO), Kyoto Encyclopedia of Genes and Genomes (KEGG), and Reactome analyses illustrating the up-regulated pathways associated with neural lineage in UCDCaP-CR cells by the bubble plot. **f** Gene Set Enrichment Analysis (GSEA) enrichment analysis showing the enrichment of neural stem cell differentiation pathways and lineage-specific markers, regulation of neuron projection development, neural lineage-associated genes, and Rickman's 966 N-Myc bivalent genes in UCDCaP-CR cells. **g** Heatmap showing both genes for the neuroendocrine (NE) feature and stem cell differentiation up-regulated in UCDCaP-CR cells. **h** The mRNA expressions (left) of N-Myc, CHGA, ENO2, and SYP in UCDCaP, UCDCaP-CR, and H660 cells. The protein expressions (right) of N-Myc, c-Myc, AR, NSE, SYP, and AURKA in H660, UCDCaP, and UCDCaP-CR cells (n = 3 samples; statistical significance determined by unpaired two-sided t-test). **i** The protein expressions of N-Myc, c-Myc, NSE, SYP, CHGA, AR, and AURKA in different prostate cancer cell lines. **j** The protein expressions of N-Myc, c-Myc, L-Myc, NSE, and SYP in adenocarcinoma (Adeno)-PDX and neuroendocrine prostate cancer (NEPC)-PDX tumors. **k** N-Myc expression in UCDCaP, UCDCaP-CR, LuCaP35CR, and LuCaP93 PDX tumors by immunohistochemical staining. Scale bar represents 20 microns. **l** Gene Expression Omnibus (GEO) datasets showing the mRNA expression of N-Myc in patients' samples (Top, n = 96 samples) and PDX samples (Bottom, n = 145 samples) (center: median; box: 25th to 75th interquartile range (IQR); whiskers: 1.5 × IQR; outliers: individual data points; statistical significance determined by one-way ANOVA with a Tukey multiple-comparison test). Results are the mean of three independent experiments ( ± S.D.). Source data are provided as a Source Data file.

(Supplementary Fig. 2c, d). Thus, the HSP70/STUB1 complex appears to be a prominent partner of N-Myc in these cells.

To understand the relationship between HSP70 and N-Myc, we performed knockdown of HSP70 in CWR22Rv1 and H660 cells, showing that chaperone depletion drastically decreased N-Myc protein but not mRNA levels (Fig. 2g and Supplementary Fig. 2e). This decrease was reversed by MG132 treatment (Fig. 2h), suggesting loss of the protein through the UPS. Interestingly, while HSC70 is among the chaperone proteins binding to N-Myc, its knockdown did not affect N-Myc protein expression (Supplementary Fig. 2f). Surprisingly, knockdown of HSC70 led to an increase in HSP70 expression, indicating distinct roles of the HSP70 family proteins in N-Myc regulation. To investigate whether STUB1 was involved in the process, we determined the mRNA and protein levels of N-Myc after overexpression of STUB1 in CWR22Rv1 and H660 cells. STUB1 overexpression significantly reduced the protein level of N-Myc (Fig. 2i) but did not affect its mRNA expression (Supplementary Fig. 2g). Similarly, we over-expressed HA-tagged N-Myc in HEK293 and C4-2B cells and observed that STUB1 markedly downregulated N-Myc protein levels in a dose-dependent manner (Supplementary Fig. 2h), suggesting that STUB1 promotes N-Myc turnover. MG132 could partially suppress this loss of either endogenous or exogenous N-Myc (Fig. 2j and Supplementary Fig. 2i), suggesting a primary role for the UPS. Moreover, in vitro ubiquitination assays demonstrated that STUB1 significantly enhanced N-Myc ubiquitination in the presence of ATP, E1, and E2 (UBE2D3) enzymes (Fig. 2k). In cells, STUB1 increased the ubiquitinated forms of N-Myc (Fig. 2l and Supplementary Fig. 2j), supporting the idea that STUB1 regulates N-Myc through the UPS. In agreement with this finding, cycloheximide chase assays revealed that overexpression of STUB1 significantly shortened, whereas knockdown of STUB1 markedly extended, the half-life of N-Myc protein (Fig. 2m, n).

In the canonical model, HSP70 acts as an adapter, in which it binds the misfolded protein and then uses a C-terminal EEVD motif to recruit STUB1 to mediate polyubiquitination[26,27]. In that model, HSP70 is thought to initially protect the client from aggregation and, if folding cannot proceed, it initiates turnover through the E3 ligase activity of STUB1[28,29]. Alternatively, STUB1 may directly interact with the native protein in a HSP70 docking-dependent mode[30]. Thus, we expected that the relative concentration of HSP70 and STUB1 or accessibility of protein domains would be important in recruiting the ligase to N-Myc, such that the proper stoichiometry or an available binding motif would be needed for ternary complex formation. Indeed, overexpression of HSP70 promoted the dissociation of STUB1 from N-Myc (Fig. 2o, p) and inhibited STUB1-mediated ubiquitination of N-Myc (Fig. 2q). The decrease in N-Myc induced by HSP70 knockdown was notably diminished with concurrent STUB1 knockdown, suggesting that the reduction in N-Myc mediated by HSP70 knockdown is primarily facilitated by STUB1 (Supplementary Fig. 2k). Moreover, knockdown of

HSP70 significantly enhanced N-Myc ubiquitination and the binding between N-Myc and STUB1 (Supplementary Fig. 2l). Taken together, these results indicate that the HSP70/STUB1 complex interacts with N-Myc and regulates N-Myc protein turnover.

## HSP70 and STUB1 directly interact with a conserved motif in N-Myc

To determine where HSP70/STUB1 binds N-Myc, we applied N-Myc mutants with different truncations to narrow down the binding region (Fig. 3a). Immunoprecipitation analysis using anti-Flag-STUB1 or anti-Flag-HSP70 antibodies revealed that HA-tagged full-length N-Myc (amino acids 1–464), N-Myc (Δ1-123), N-Myc (Δ382-464), and N-Myc (Δ346-464)[31] were detectable in the immunoprecipitation containing Flag-STUB1 and Flag-HSP70 (Fig. 3b, c). However, N-Myc (Δ281-464) did not immunoprecipitate with the chaperones, indicating that the region in N-Myc containing amino acid residues 281–345 is important for binding to STUB1 and HSP70. In this region, we noted a sequence $^{316}$SELILKR$^{322}$ that resembles a classical HSP70-binding motif[32], corresponding to hydrophobic amino acids followed by cationic ones (Fig. 3d). To test this prediction, we constructed truncations of N-Myc that deleted this region (Δ314-342) or a subset of this sequence (ΔLILKR) (Fig. 3e). We also created a mutation replacing this sequence with the poor HSP70-binding sequence, CLPQS (Supplementary Fig. 3a). In co-immunoprecipitation assays, all three constructs (N-Myc Δ314-342/ΔLILKR/CLPQS) bound poorly to HSP70 and STUB1, compared to wild-type N-Myc (Fig. 3f and Supplementary Fig, 3b-c). These results suggest that either HSP70 binds this site and recruits STUB1 or that HSP70 and STUB1 share this binding motif. To explore these interactions using an independent assay, we performed proximity ligation assays (PLAs). In those experiments, expression of N-Myc ΔLILKR reduced the proximity with HSP70 and STUB1 compared to that of WT-N-Myc (Fig. 3g), consistent with disruption of the complex. However, both WT-N-Myc and N-Myc ΔLILKR remain interact with MAX (Supplementary Fig. 3d). Moreover, the half-life of ΔLILKR and CLPQS mutants was significantly extended compared to that of wild-type N-Myc (Fig. 3h and Supplementary Fig. 3e). Overexpression of STUB1 had no effect on levels and ubiquitination of the N-Myc ΔLILKR and CLPQS mutants (Fig. 3i, j and Supplementary Fig. 3f). Thus, the 'SELILKR' motif appears to be the key sequence for HSP70/STUB1 binding on N-Myc.

STUB1 consists of a coiled-coil domain, a tetratricopeptide repeat (TPR) domain, and a Ubox domain (Supplementary Fig. 3g). Mutation of a single residue in the TPR domain, K30A, is known to disrupt binding to HSP70 by disrupting a conserved "carboxylate clamp". Likewise, a H260Q mutation in the Ubox domain abolishes its E3 ligase activity[33]. Using these mutations, plus a series of truncations, we asked which interactions and activities are required for STUB1 binding to HSP70 and N-Myc. Using HEK293 cells, we co-transfected HSP70 or N-Myc with various STUB1 constructs (wild-type, K30A, ΔTPR, H260Q,

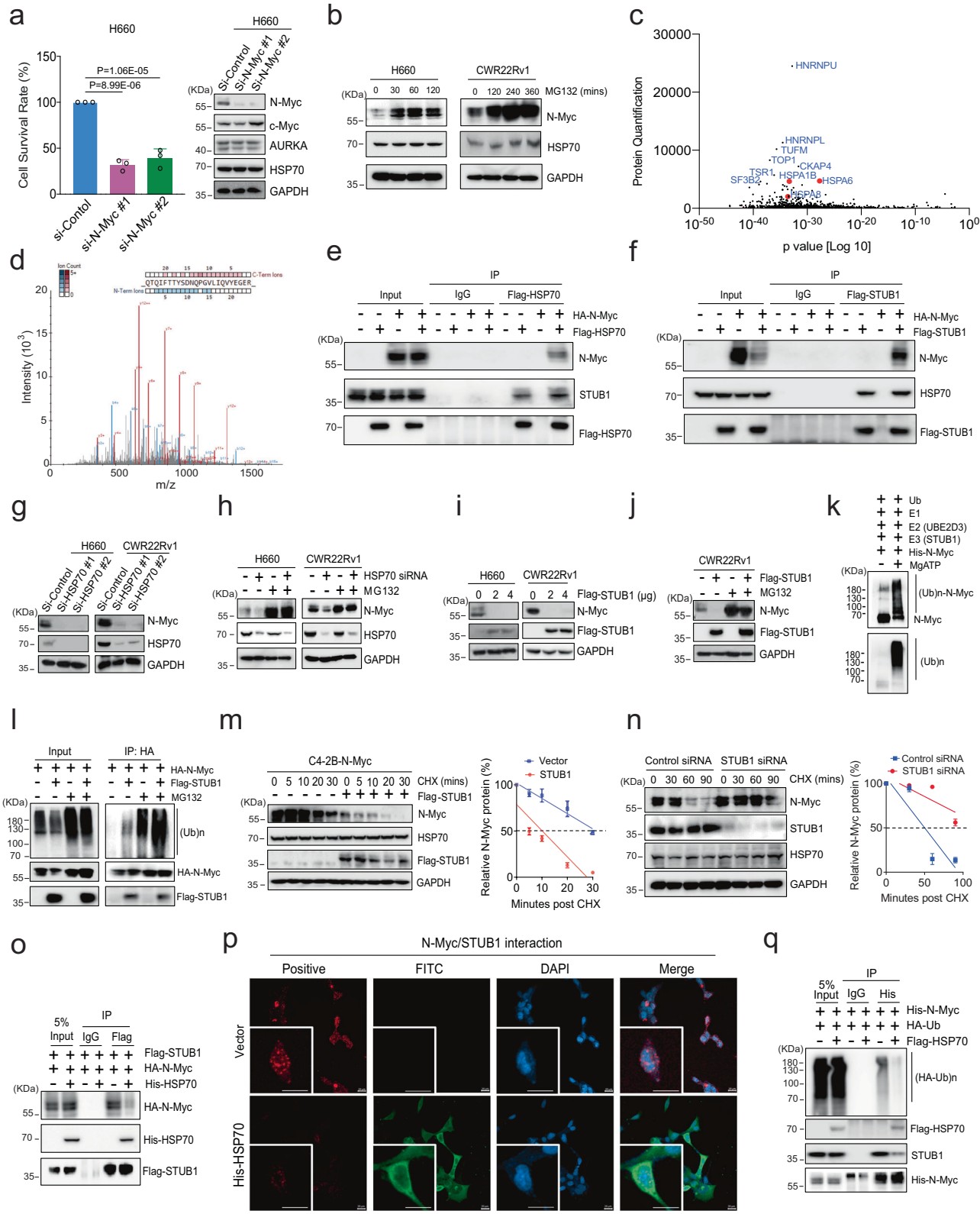

ΔUBox) and performed immunoprecipitation assays using anti-HSP70 or anti-HA antibodies. We found that deletion of STUB1's TPR domain blocked its interactions with both HSP70 and N-Myc. However, the STUB1 K30A mutant only lost its binding to HSP70, but not N-Myc (Supplementary Fig. 3h), suggesting that the carboxylate clamp of the TPR domain is not required for binding N-Myc. Thus, STUB1 seems to have at least two ways of interacting with N-Myc, through HSP70 and a chaperone-independent interaction that involves its TPR domain which is consistent with other proteins that have recently been found to engage with STUB1 in a chaperone-independent way[34]. In the case of N-Myc, this interaction seems to occur at a degron 'SELILKR' in N-Myc.

**Fig. 2 | HSP70/STUB1 controls N-Myc protein levels through the ubiquitin proteasome system. a** Cell proliferation and N-Myc expression were assessed in H660 cells transfected with control and N-Myc siRNA (n = 3; statistical significance determined by one-way ANOVA with a Tukey multiple-comparison test). **b** H660 and CWR22Rv1 cells were cultured with MG132 for different times. N-Myc expression was examined by western blotting. **c** N-Myc binding proteins were determined by the immunoprecipitation pull-down assay followed by proteomic profiling. **d** The representative HSPA1B peptide pull-down with N-Myc was listed. **e, f** HEK293 cells were co-transfected with N-Myc with or without Flag-HSP70 or Flag-STUB1, and the cell lysates were immunoprecipitated with the anti-Flag antibody. **g, h** H660 and CWR22Rv1 cells were transfected with control and HSP70 siRNA, or then treated with or without MG132 for 6 h. Whole cell lysates were separated by electrophoresis and blotted for N-Myc and HSP70. **i, j** The protein expression of N-Myc was determined after H660 and CWR22Rv1 cells were transfected with Flag-STUB1, or then treated with or without MG132 for 6 h. **k** In vitro ubiquitination assays were conducted to detect the ubiquitination of N-Myc. **l** HEK293 cells were transfected with HA-N-Myc or with Flag-STUB1 and treated with or without MG132 for 6 h. Immunoprecipitation was performed with the anti-HA antibody. **m** C4-2B cells overexpressing N-Myc were transfected with control or Flag-STUB1 and treated with cycloheximide for different times. N-Myc expression was analyzed by western blotting to calculate its half-life (n = 3 independent experiments; data are presented as mean ± S.D.). **n** Similar to (**m**), but CWR22Rv1 cells were transfected with STUB1 siRNA or control before cycloheximide treatment (n = 3 independent experiments; data are presented as mean ± S.D.). **o** Lysates of HEK293 cells co-transfected with HA-N-Myc, Flag-STUB1, and His-HSP70 were immunoprecipitated with the anti-Flag antibody. **p** HEK293 cells were co-transfected with HA-N-Myc and Flag-STUB1 with or without His-HSP70, and the interaction of N-Myc and STUB1 was determined by Proximity Ligation Assay (PLA). His-HSP70 was stained by fluorescein isothiocyanate (FITC). Scale bar represents 40 microns. **q** HEK293 cells were co-transfected with His-N-Myc and HA-Ubiquitin with or without Flag-HSP70, and the whole cell lysates were immunoprecipitated with the anti-His antibody. Results are the mean of three independent experiments ( ± S.D.). Source data are provided as a Source Data file.

## STUB1 polyubiquitinates a specific region of N-Myc to mediate its degradation

Next, we explored which sites on N-Myc might be subject to modification by STUB1. First, we co-transfected STUB1 K30A, ΔTPR, H260Q, and ΔUBox with N-Myc in HEK293 cells and performed western blotting and co-IP analysis. Only wild-type STUB1 significantly reduced N-Myc levels (Fig. 4a) and promoted N-Myc ubiquitination (Fig. 4b). The inability of K30A STUB1 to mediate N-Myc turnover suggests that, in this case, the HSP70-STUB1 complex is more efficient at polyubiquitination. Having established that using wild type STUB1 is critical in these experiments, we examined its effect on the levels of truncated forms of N-Myc, revealing that the 382–464 domain (BR-HLH-LZ domain) was the most crucial (Fig. 4c and Supplementary Fig. 4a). Consistently, deletion of 382–464 inhibited the STUB1-mediated ubiquitination of N-Myc (Fig. 4d). Furthermore, the half-life of Δ382-464 N-Myc was 3-fold extended compared to that of wild-type N-Myc (60 min vs. 20 min) (Fig. 4e). Together, these experiments suggest that HSP70/STUB1 complex positions the ligase to act on lysine residues in the 382–464 region. There are nine lysine residues in the 382–464 domain of N-Myc (K413, K416, K419, K424, K425, K444, K446, K456, and K457) (Fig. 4f). At each site, we made arginine replacement, using K52R as a negative control (Supplementary Fig. 4b). In HEK293 cells co-transfected with STUB1, the levels of the K416R and K419R mutants appeared most resistant to turnover (Fig. 4g). Notably, STUB1-mediated ubiquitination of N-Myc was also reduced in these two mutants compared to WT-N-Myc (Fig. 4h). Two negative control sites (K52R and K413R) retained the ubiquitination status as that of WT-N-Myc (Supplementary Fig. 4c), suggesting that K416 and K419 are the key STUB1-mediated ubiquitination sites.

Polyubiquitin chains can be formed from successive modification of ubiquitin (Ub) on seven lysines (K6, K11, K27, K29, K33, K48, and K63) and the type of chain linkage is thought to, in part, guide the fate of the modified protein. To dissect which types of chains are added to N-Myc by STUB1, we co-transfected each of the seven Ub mutants (K6R, K11R, K27R, K29R, K33R, K48R, and K63R) and wild-type Ub with N-Myc and STUB1 into HEK293 cells. These experiments revealed that K11R- and K63R- partially blocked the activity of STUB1, suggesting that the ligase prefers to use these linkages on N-Myc (Fig. 4i). As a control, we found that Ub-K0 with all seven lysine sites mutated significantly blocked STUB1-mediated ubiquitination of N-Myc (Supplementary Fig. 4d). However, reversion of K11 or K63 in the Ub-K0 construct showed the restoration of polyubiquitination of N-Myc (Fig. 4j and Supplementary Fig. 4e). Notably, only K11-linked but not K63-linked N-Myc ubiquitination can be regulated by the MG132 treatment (Fig. 4k), suggesting that the proteasome mediated N-Myc degradation is through the K11 site.

Thus, K416 and K419 seem to be the sites of ubiquitin modification by STUB1, where it adds K11-linked polyubiquitin chains for further proteasome degradation.

## Pharmacologically targeting HSP70 degrades N-Myc expression via STUB1

The results thus far suggest a model in which HSP70 binds N-Myc and works together with STUB1 to mediate degradation through the UPS. The dwell time of HSP70 on its clients is thought to be a key determinant of whether STUB1 is recruited[35]. Thus, we predicted that allosteric inhibition of HSP70's ATPase activity would "stall" the chaperone on N-Myc and promote its turnover. The chemical probe, JG231, was designed to stabilize the ADP-bound state of HSP70 that has tight affinity for clients[36,37]. To determine whether JG231 affects N-Myc expression, we first tested its impact on the endogenous and exogenous protein in CWR22Rv1, H660, UCDCaP-CR, HEK293, and N-Myc-overexpressing C4-2B cells. We found that treatment induced a significant and dose-dependent reduction in N-Myc levels in all the cell lines tested. HSP70 protein level was slightly decreased in high dose treatment (Fig. 5a, b and Supplementary Fig. 5a, b). As expected, mRNA levels of MYCN were unaffected (Supplementary Fig. 5c). Furthermore, N-Myc inhibition by JG231 was partially reversed by MG132 treatment, suggesting that JG231 reduces N-Myc expression through the UPS (Fig. 5c, d). JG231 significantly shortened the half-life of N-Myc (Fig. 5e). In the presence of MG132, JG231 treatment notably elevated global protein ubiquitination. Moreover, this treatment augmented the duration of the HSP70/STUB1 complex interaction with N-Myc, consequently enhancing its ubiquitination (Fig. 5f). Strikingly, STUB1 knockdown abrogated the inhibitory effect of JG231 on N-Myc, indicating that JG231-induced degradation of N-Myc protein occurs primarily through STUB1 (Fig. 5g). Thus, allosteric inhibition of HSP70 seems to promote STUB1-mediated degradation of N-Myc.

HSP70 and STUB1 are typically thought to reside in both the cytoplasm and nuclei, whereas N-Myc is a primarily nuclear protein. Accordingly, we wanted to explore whether treatment with JG231 might alter the relative distribution of these proteins in subcellular locations. As expected, we found that N-Myc was localized to the nuclei, whereas both STUB1 and HSP70 were present in both the cytoplasm and nuclei. JG231 treatment reduced the levels of N-Myc and HSP70 in the nuclei, but, interestingly, seemed to increase STUB1 levels (Supplementary Fig. 5d). From the dual immunofluorescence and PLA images, we found that JG231 treatment promoted STUB1 entry into the nuclei and its proximity to N-Myc (Fig. 5h, i). Further co-IP assays confirmed that JG231 promoted N-Myc and STUB1 binding under these conditions (Supplementary Fig. 5e) and increased N-Myc ubiquitination in the nuclei (Fig. 5j). Treatment with JG231 had no effect on N-Myc ΔLILKR, confirming that the HSP70-binding site is

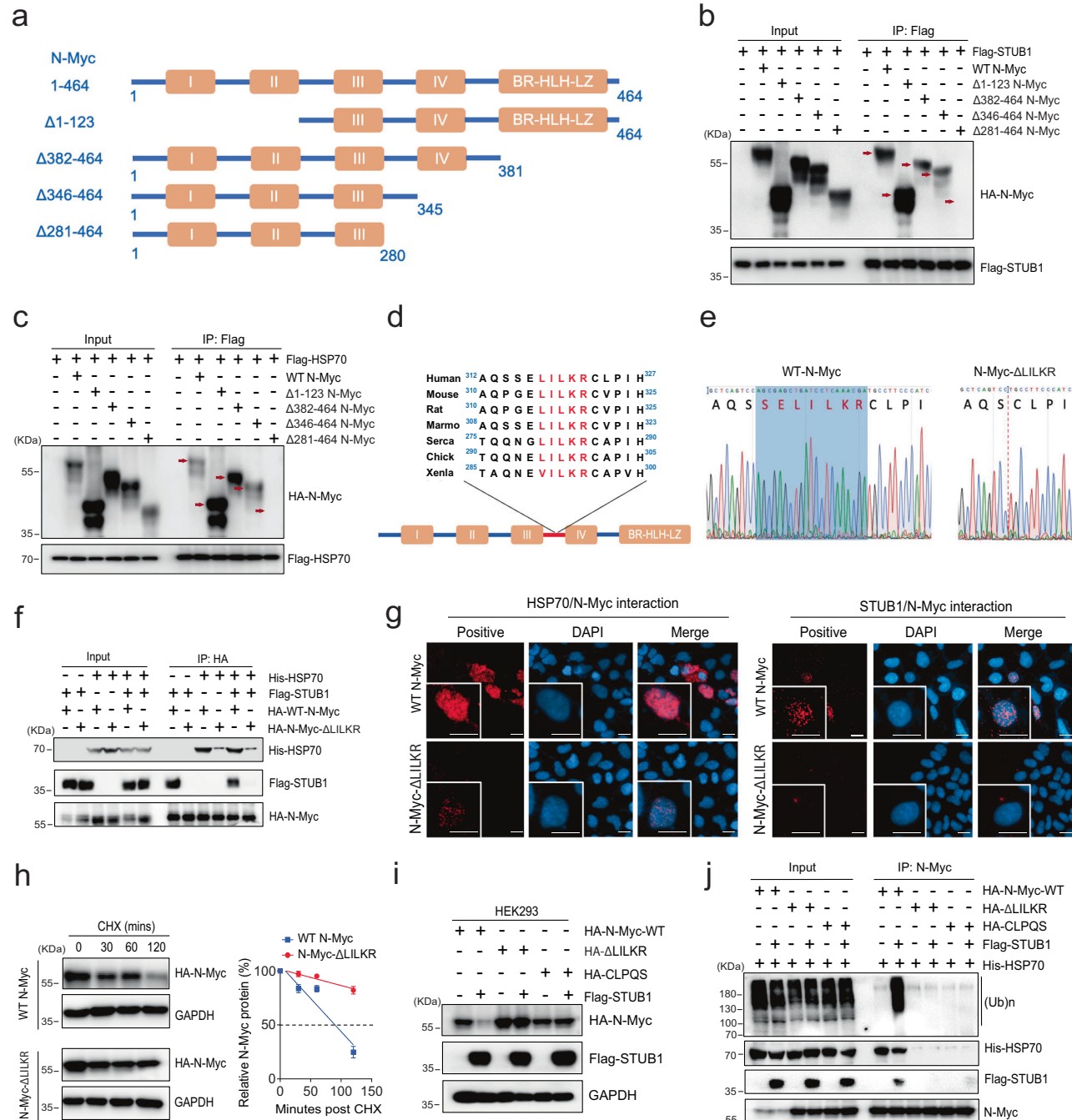

**Fig. 3 | HSP70 in interplay with STUB1 directly interacts with an N-Myc domain.**
**a** Schematic representation of the N-Myc deletion mutants used for domain mapping. Myc boxes: I, II, III, IV; BR, basic region; HLH, helix-loop-helix; LZ, leucine zipper. **b** HEK293 cells were co-transfected with Flag-STUB1 and wild-type or indicated HA-N-Myc mutant constructs. The cell lysates were immunoprecipitated with the anti-Flag antibody. The red arrows mark the expected positions of the full-length or truncated N-Myc pulled down by STUB1. **c** HEK293 cells were co-transfected with Flag-HSP70 and wild-type or indicated HA-N-Myc mutant constructs. The cell lysates were immunoprecipitated with the anti-Flag antibody. The red arrows mark the expected positions of the full-length or truncated N-Myc pulled down by HSP70. **d** Alignment of the potential binding sites in N-Myc in different species to HSP70. **e** The Sanger Sequence chromatogram of the wild-type (WT) N-Myc plasmid and the corresponding deletion in N-Myc-ΔLILKR. **f** HEK293 cells co-transfected with/without His-HSP70, Flag-STUB1, and HA-WT-N-Myc or HA-N-Myc-ΔLILKR. Whole cell lysates were harvested and immunoprecipitated with the

anti-HA antibody. **g** HEK293 cells were co-transfected with HA-WT-N-Myc or HA-N-Myc-ΔLILKR, with Flag-HSP70 or Flag-STUB1 for 3 days, and the interaction of N-Myc and HSP70 or STUB1 was determined by Proximity Ligation Assay (PLA). Scale bar represents 20 microns. **h** HEK293 cells were transfected with HA-WT-N-Myc or HA-N-Myc-ΔLILKR and treated with 50 µg/ml cycloheximide for 0, 30, 60, and 120 min. Whole cell lysates were separated by electrophoresis and blotted with the anti-HA antibody, and the half-life of the full-length and deleted N-Myc molecules was calculated (n = 3 independent experiments and data presented as mean ± S.D.). **i** HEK293 cells were transfected with HA-WT-N-Myc, HA-N-Myc-ΔLILKR, or HA-N-Myc-CLPQS, with or without Flag-STUB1 for 3 days. Total cell lysates were collected for western blotting to detect the expression of N-Myc. **j** HEK293 cells co-transfected with His-HSP70, HA-WT-N-Myc, HA-N-Myc-ΔLILKR, or HA-N-Myc-CLPQS and Flag-STUB1 plasmids. Whole cell lysates were immunoprecipitated with the anti-N-Myc antibody. Source data are provided as a Source Data file.

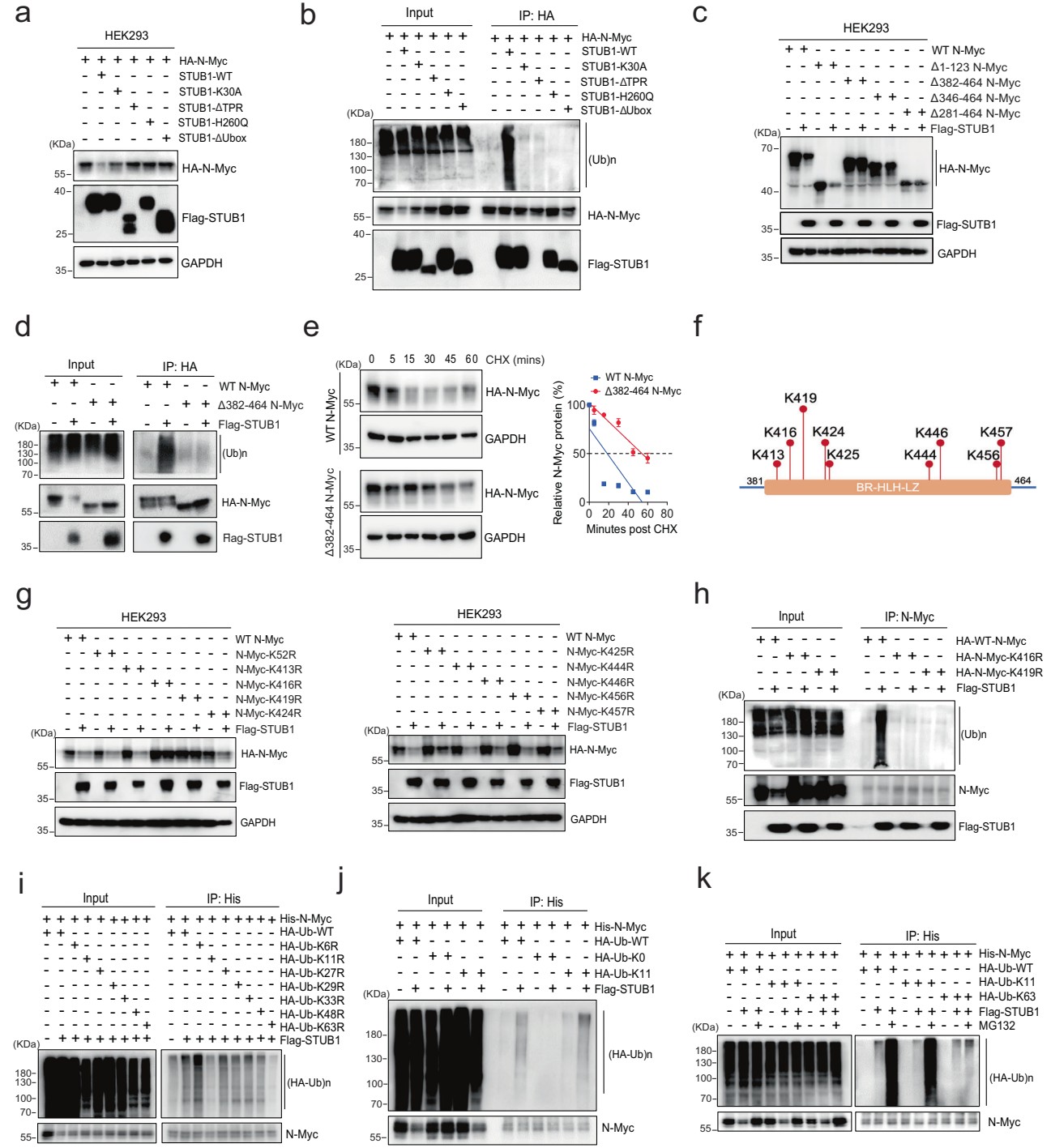

required (Supplementary Fig. 5f). JG231 also demonstrated limited effects on the STUB1 ubiquitination-dead mutants (N-Myc K416R and K419R) (Supplementary Fig. 5g). Co-IP assays further confirmed that JG231 led to reduced ubiquitination on these mutants (Fig. 5k). Furthermore, the JG231 treatment was found to promote ubiquitin-linked polyubiquitination chains specifically on K11 and K63 sites (Fig. 5l). This observation strongly suggests that JG231 acts through HSP70/STUB1 complex, and not an off-target, to control N-Myc levels in this system. Functionally, WT-N-Myc partially restored cell viability in UCDCaP-CR following JG231 treatment, whereas the two STUB1 ubiquitination-dead mutants exhibited even greater rescue effects compared to WT-N-Myc (Supplementary Fig. 5h). These findings underscore the pivotal role of N-Myc in regulating cell proliferation in NEPC cells and suggest that

JG231 effects are partially mediated through N-Myc protein regulation. N-Myc typically forms a complex with MAX to initiate downstream gene expression[38]. We found that treatment with JG231 led to the dissociation of N-Myc from MAX by co-IP (Supplementary Fig. 5i) or by PLA (Supplementary Fig. 5j). Thus, HSP70 binding seems to antagonize the active N-Myc/MAX complex and stabilization of HSP70 binding by JG231 favors the inactive N-Myc state.

To comprehensively understand the broad and systemic impact of JG231 on the cellular proteome, we conducted proteomic profiling using LC/MS in C4-2B N-Myc cells treated with JG231. The overexpression of N-Myc induced the elevation of NE markers (NSE and SYP), suggesting that N-Myc plays a functional role in transforming CRPC to NEPC (Supplementary Fig. 5k). Subsequently, C4-2B N-Myc

**Fig. 4 | STUB1 coordinates with HSP70 in regulating N-Myc polyubiquitination and degradation. a** HEK293 cells were co-transfected with wild-type (WT) STUB1 or mutants and HA-N-Myc. The expression of N-Myc was determined by western blotting. **b** HEK293 cells were co-transfected with Flag-WT-STUB1 or mutants and HA-N-Myc. Whole cell lysates were immunoprecipitated with the anti-HA antibody and blotted with anti-ubiquitin, Flag, or HA antibodies. **c** HEK293 cells were co-transfected with HA-WT-N-Myc or mutants in the absence or presence of the Flag-STUB1 construct. The expression of N-Myc was determined by western blotting. **d** HEK293 cells were co-transfected with HA-WT-N-Myc or Δ382-464-N-Myc and Flag-STUB1 plasmids. Whole cell lysates were immunoprecipitated with the anti-HA antibody and blotted with Ubiquitin, Flag, or HA antibodies. **e** HEK293 cells were transfected with HA-WT-N-Myc or Δ382-464-N-Myc for 3 days and then treated with 50 μg/mL cycloheximide. Total cell lysates were collected at 0, 5, 15, 30, 45, and 60 min after the treatment and subjected to western blotting. Half-lives of WT and truncated N-Myc were calculated (n = 3 independent experiments, data presented as mean ± S.D.). **f** Schematic representation of potential ubiquitination modification sites within the residues 382–464 in N-Myc. **g** HEK293 cells were co-transfected with HA-N-Myc WT or mutants and Flag-STUB1 plasmids. The expression of N-Myc was determined by western blotting. **h** HEK293 cells were co-transfected with HA-WT-N-Myc or selected mutants and Flag-STUB1 plasmids. Whole cell lysates were immunoprecipitated with the anti-N-Myc antibody and blotted with anti-ubiquitin, Flag, or N-Myc antibodies. **i** HEK293 cells were co-transfected with His-N-Myc, HA-WT-Ubiquitin, or Ub mutants and Flag-STUB1. Whole cell lysates were immunoprecipitated with the His-Tag Dynabeads™ and blotted with the HA or N-Myc antibodies. **j** HEK293 cells were co-transfected with His-N-Myc, HA-Ubiquitin (WT, K0, or K11) and Flag-STUB1 plasmids. Cell lysates were immunoprecipitated with the His-Tag Dynabeads™ and probed for HA or N-Myc. **k** HEK293 cells were co-transfected with His-N-Myc, HA-Ubiquitin (WT, K11, or K63), and Flag-STUB1 plasmids with or without 5 μM MG132. Cell lysates were immunoprecipitated with the His-Tag Dynabeads™ and probed for HA or N-Myc. Source data are provided as a Source Data file.

cells were treated with DMSO, 2.5, or 5 μM JG231 for 4 h. As confirmed by western blot analysis, 5 μM JG231 reduced N-Myc expression by 90% within 4 h of treatment (Supplementary Fig. 5l). Further analysis of the nuclear fraction confirmed that JG231 treatment significantly degraded N-Myc protein expression within a very short time frame (Supplementary Fig. 5m). Whole cell lysates and nuclear lysates were then subjected to LC/MS for DIA profiling. In whole cell lysates, a total of 3808 proteins were detected. Among these, 494 proteins exhibited decreased levels, while 162 proteins showed increased levels following JG231 treatment (Supplementary Fig. 5n). However, in the nuclear lysates, a total of 5524 proteins were detected. Among these, 917 proteins exhibited decreased levels, including N-Myc, c-Myc, AR, USP36, USP39, and USP42 while 416 proteins showed increased levels following JG231 treatment, including UBB, HABP2, TIMM9, and LZTR1 (Fig. 5m). Notably, only the nuclear fraction was able to detect N-Myc protein by LC/MS, possibly due to its rapid turnover and stability issues. Among the downregulated proteins in whole cell lysates, significant alterations were observed in protein stabilization, protein folding, protein binding, ribosome, and mitochondrial function (Supplementary Fig. 5o, p, Supplementary Data 1). However, among the downregulated proteins in nuclear lysates, significant alterations were observed in RNA splicing, protein localization, protein binding, RNA binding, ATPase activity, ribosome, cell cycle, axon guidance, and peptide chain elongation (Fig. 5n, o, Supplementary Data 2).

## HSP70 inhibitor JG231 suppresses NEPC xenograft tumor growth and neuroendocrine signatures

We predicted that loss of N-Myc after JG231 treatment might suppress growth of NEPC cells. To test this idea, three NEPC- or NE-like cell lines, H660, CWR22Rv1, and UCDCaP-CR, were treated, and proliferation measured by cell counting and the CellTiter-Glo Luminescent assay. In these experiments, JG231 inhibited the proliferation of NEPC cells in a dose-dependent manner. Normal fibroblast cell line IMR90 and immortalized prostate epithelial cell line RWPE-1 were also tested. Notably, both IMR90 and RWPE-1 cells exhibit lower sensitivity to JG231 treatment compared to cancer cells, with IC50 values approximately 5- to 10-fold higher (Fig. 6a). JG231 also induced the death of NEPC PDX LuCaP93 organoids in a dose-dependent manner (Fig. 6b). In addition, we implanted the LuCaP93 PDX tumor model and evaluated the therapeutic effect of JG231 in vivo. JG231 treatment significantly inhibited tumor growth and tumor weight (Fig. 6c, d). Immunohistochemical staining for N-Myc and Ki67 showed that JG231 treatment significantly reduced N-Myc levels and tumor cell proliferation (Fig. 6e). This finding was confirmed in another NEPC xenograft model, where JG231 treatment significantly slowed H660 tumor growth (Fig. 6f), and decreased N-Myc and Ki67 expression (Fig. 6g). It is noteworthy that we observed no significant alteration in global protein polyubiquitination following JG231 treatment in both LuCaP93

and H660 tumors (Supplementary Fig. 6a). This could suggest that the UPS and the autophagy system are involved in maintaining the cellular ubiquitination status throughout the treatment period.

HSP70 plays many important housekeeping functions; however, cancer cells seem to have a strong reliance on its function that creates a therapeutic window for treatment with allosteric HSP70 inhibitors[39–41]. Consistent with this idea, we found that JG231 treatment did not affect the body weight (Supplementary Fig. 6b) or gross morphology of the organs in mice (Supplementary Fig. 6c). Moreover, no significant pathological changes were observed after JG231 treatment: livers did not show any vacuolar changes, and there was no sign of inflammation in the renal pelvis after JG231 treatment (Fig. 6h).

To understand the effects of JG231 on gene expression in NEPC cells, we treated H660 cells with JG231 and performed RNA sequencing. Gene expression profiles were significantly altered in JG231-treated cells compared to those in control cells. GO, KEGG, and Reactome dot analyses and GSEA showed that multiple pathways related to the cell proliferation, regulation of translation, ribosome, apoptosis, P53 pathway, cell cycle, and unfolded protein response (UPR) were significantly upregulated in JG231-treated cells (Fig. 6i and Supplementary Fig. 6d, e). These results are consistent with recent findings in other cancer cells[42,43]. However, we also noted that signaling pathways related to neuron development and neurotransmitters, such as synaptic signaling, positive regulation of nervous system development, neuroactive ligand-receptor interaction, synapse assembly, axoneme assembly, and regulation of synapse assembly were significantly downregulated (Fig. 6j and Supplementary Fig. 6f, g). We also analyzed Rickman's 966 N-Myc Bivalent Genes dataset[17] and found that the expression of these genes was significantly downregulated after JG231 treatment (Fig. 6k). Classical NE markers, such as SYP, ENO2, CHGA, and NCAM1, were significantly suppressed by the JG231 treatment (Supplementary Fig. 6h, i), suggesting the potential reversal of NE features. Taken together, these results suggest that JG231 treatment suppresses NEPC cell lines, organoids, and xenograft tumor growth, and regulates gene expression in NEPC cells.

## Dual targeting of HSP70 and AURKA enhances N-Myc degradation and improves treatment in NEPC

During the bioinformatics analysis of the JG231 treated H660 dataset, we found that the signaling pathway related to AURKA was activated after JG231 treatment (Supplementary Fig. 7a). JG231 treatment did not disassociate the binding of AURKA and N-Myc (Supplementary Fig. 7b), suggesting a potential bypass route present in the cancer cells. It is well known that AURKA interacts with N-Myc and protects it from degradation by the proteasome system[44]. The AURKA inhibitor alisertib inhibits the growth of NEPC tumors by disrupting the N-Myc signaling pathway; however, it failed in a phase II clinical trial[45]. Therefore, we reasoned that a combination of JG231 and alisertib might be interesting

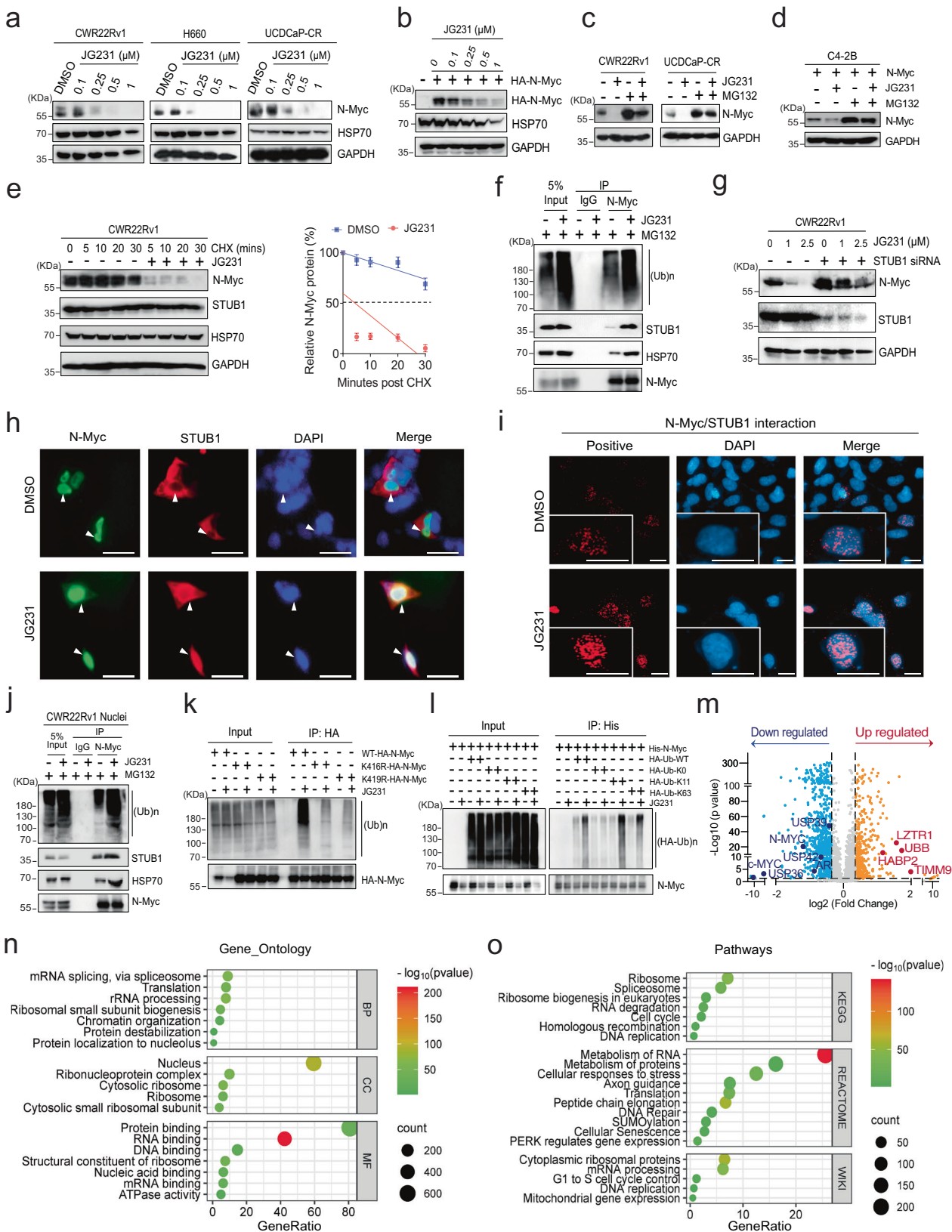

as a treatment for NEPC. Proof-of-principle studies confirmed that either HSP70 siRNA or JG231 treatment significantly improved AURKA siRNA effects in growth inhibition of CWR22Rv1 cells (Fig. 7a, b). AURKA knockdown suppressed N-Myc protein expression in NEPC cells (Supplementary Fig. 7c). However, JG231 treatment further

improved the effects of AURKA siRNA on N-Myc suppression (Fig. 7c). Treatment with alisertib suppressed the proliferation of CWRR22Rv1, H660, and UCDCaP-CR cells in a dose-dependent manner (Fig. 7d). However, its ability to reduce the levels of N-Myc protein are marginal. Strikingly, JG231 drastically improved the effects of alisertib on N-Myc

**Fig. 5 | Pharmacologically targeting HSP70 degrades N-Myc expression via STUB1. a** CWR22Rv1, H660, and UCDCaP-CR cells were treated with JG231, and N-Myc expression was determined by western blotting. **b** Similar to a, but HEK293 cells were transfected with HA-N-Myc for 2 days before JG231 treatment. **c** CWR22Rv1 and UCDCaP-CR cells were treated with JG231 (2.5 μM) and MG132. The expression of N-Myc was determined by western blotting. **d** Similar to c, but C4-2B cells were transfected with N-Myc before JG231 treatment. **e** CWR22Rv1 cells were treated with cycloheximide in the absence or presence of JG231 (10 μM). N-Myc expression was analyzed by western blotting to calculate its half-life (n = 3 independent experiments, data presented as mean ± S.D.). **f** CWR22Rv1 cells were treated with JG231 (5 μM) overnight and then treated with MG132, and immunoprecipitation was performed with N-Myc antibody. **g** CWR22Rv1 cells were transiently transfected with STUB1 siRNA, followed by treatment with JG231. The cell lysates were collected and subjected to western blotting. **h, i** HEK293 cells were co-transfected with HA-N-Myc and Flag-STUB1, followed by treatment with JG231

(2.5 μM) and MG132. The localization and interaction of N-Myc and STUB1 were analyzed by immunofluorescence and Proximity Ligation Assay (PLA), respectively. Scale bars represent 50 and 20 microns, respectively. **j** CWR22Rv1 cells were treated with JG231 (2.5 μM) and then treated with MG132. The nuclear protein was immunoprecipitated with N-Myc antibody. **k** HEK293 cells were transfected with WT-N-Myc, N-Myc-K416R, or N-Myc-K419R, and then treated with JG231 (2.5 μM). Immunoprecipitation was performed with HA antibody. **l** HEK293 cells were co-transfected with His-N-Myc, HA-Ubiquitin (WT, K0, K11, or K63), and then treated with JG231 (5 μM). Immunoprecipitation was performed with His-tag Dynabeads™. **m** The volcano plot shows the nuclear proteins from C4-2B N-Myc cells treated with JG231 for 4 h by proteomic profiling. Blue dots represent down-regulated, and yellow dots represent up-regulated (1.3-fold and p < 0.05) proteins. **n, o** Gene Ontology and pathway analyses demonstrate the enrichment of functional annotations in down-regulated proteins in nuclear lysates from C4-2B N-Myc cells treated with 5 μM JG231. Source data are provided as a Source Data file.

(Fig. 7e). Together, these finding support the idea that AURKA and HSP70 are important mediators of N-Myc proteostasis in NEPC cells.

We subsequently investigated the combined effect of alisertib and JG231 on the proliferation of prostate cancer cells overexpressing N-Myc. Compared to the control and single treatment groups, the combined treatment significantly inhibited cell proliferation, demonstrating synergistic effects (Fig. 7f and Supplementary Fig. 7d). These findings were confirmed using colony formation assays, where alisertib and JG231 suppressed colony formation in a dose-dependent manner, and combination treatment further reduced colony size and number (Fig. 7g). Notably, these combination effects were also observed if we replaced alisertib with other, chemically distinct AURKA inhibitors. Specifically, the results of experiments using three additional AURKA inhibitors (MK-8745, AURKA inhibitor, and tozasertib) were similar to those of alisertib (Supplementary Fig. 7e, f). This combination treatments were also tested in NEPC organoids. Combined treatment of JG231 cells with alisertib or the three other AURKA inhibitors significantly induced the death of LuCaP93 and H660 organoids (Fig. 7h and Supplementary Fig. 7g, h).

To further determine whether JG231 enhances alisertib treatment efficacy in vivo, we performed animal experiments using an H660 xenograft model. The combination of JG231 and alisertib further inhibited tumor growth compared to the single treatment groups, especially in the alisertib treatment alone group (Fig. 7i). The toxicity panel data showed that neither single treatment with JG231 nor the combination treatment altered kidney and liver function (Fig. 7j, k and Supplementary Fig. 7i). Immunohistochemical staining of N-Myc and Ki67 illustrated that N-Myc expression and cell proliferation were slightly inhibited by JG231, or alisertib treatment alone but profoundly inhibited by the combination treatment (Fig. 7l). Collectively, JG231 and alisertib synergistically suppressed N-Myc expression as well as the growth of NEPC cell lines, organoids, and xenograft tumors.

## Discussion

Components of the proteostasis network, including the molecular chaperones, the ubiquitin-proteasome, and the autophagy systems play important roles in tumorigenesis, drug resistance, and cancer progression[24,46–48]. Sustained proteostasis can maintain oncogenic pathways that enhance cancer cell survival and progression, especially in the context of applied therapies[4,49–54]. To better understand these mechanisms, we focus here on the amplification of N-Myc, which is associated with prostate cancer when the tumor lineage switches from epithelial to neuroendocrine[17,18]. N-Myc together with the other two oncogenes c-Myc and L-Myc are members of the Myc family which has been connected to cancer development and progression[55,56]. Because of the intrinsically disordered functional domains and lack of distinct binding pockets on the proteins, Myc proteins are considered undruggable. Using proteomics and biochemical studies, we found

that HSP70 is one of the chaperone proteins of N-Myc that can regulate its proteostasis. Evidence has shown that HSP70 and STUB1 play roles in multiple protein regulation in the cancer setting[29,57]. AR-V7 is one of the substrate proteins regulated by HSP70/STUB1 complex in enzalutamide resistant CRPC[39,41]. Here we have identified a broader application where HSP70 works together with STUB1 in controlling N-Myc turnover in NEPC. In addition, we have provided significantly mechanistic insight into this process as illustrated in Fig. 8. Specifically, we identified a conservative motif, "SELILKR", that binds to HSP70 and STUB1. This binding motif seems to serve as a degron because HSP70 can recruit active STUB1 and promote misfolded N-Myc ubiquitination and degradation. However, when HSP70 is up-regulated as in advanced prostate cancer, it occupies the "SELILKR" binding motif on native N-Myc protein and prevent the access of STUB1 possibly through steric hindrance. The turnover of N-Myc becomes consequently slow. With the presence of JG231, HSP70 is converted to the ADP-bound state, stalls, and holds STUB1 in close proximity to N-Myc. The result of this ternary complex formation is that N-Myc is ubiquitinated on K416 and K419 sites and formed K11-linked polyubiquitinated chains for UPS degradation. Based on our findings, N-Myc also undergoes K63-linked polyubiquitination which will possibly direct the protein to form autophagosome for degradation[58]. Interestingly, unlike the stress-inducible HSP70, the constitutive HSP70 isoform, HSC70, is unable to regulate N-Myc turnover. Despite their 85% similarity, these proteins seem play distinct roles in N-Myc protein regulation, warranting further investigation.

There has been a long-standing interest in targeting Myc for cancer treatments. For example, many researchers have synthesized small molecules to disrupt Myc binding to MAX or its binding to DNA. Compounds such as MYCi361 and its improved derivative MYCi975[59], L755507[60] interfere with the Myc-MAX interaction, and VPC-70619 was developed to selectively target N-Myc-Max binding. However, we suggest that it is interesting to consider disruption of N-Myc proteostasis as a viable alternative. Here, we used an allosteric inhibitor of HSP70, JG231, to promote N-Myc degradation in cells, organoid, and animal models. We found that the gene expression profile of JG231-treated cells suggested that AURKA signaling was possibly affected. This is interesting because concurrent amplification of AURKA and N-Myc has been detected in t-NEPC, and evidence shows that the catalytic domain of AURKA directly binds to c-Myc around threonine-58, which is also present in N-Myc[61]. These findings led us to try combinations of the AURKA inhibitor, alisertib, with JG231, revealing a striking synergy in these compounds. A phase II clinical trial using alisertib was conducted in 60 CRPC/NEPC patients. The outcome did not meet the primary endpoint, however, the molecular relations between N-Myc and AURKA was validated, and alisertib remains as a potential single agent to treat NEPC[45,62]. Given the relatively poor activity of alisertib as a single agent, the combination of both HSP70 and AURKA inhibitors might be worth pursuing.

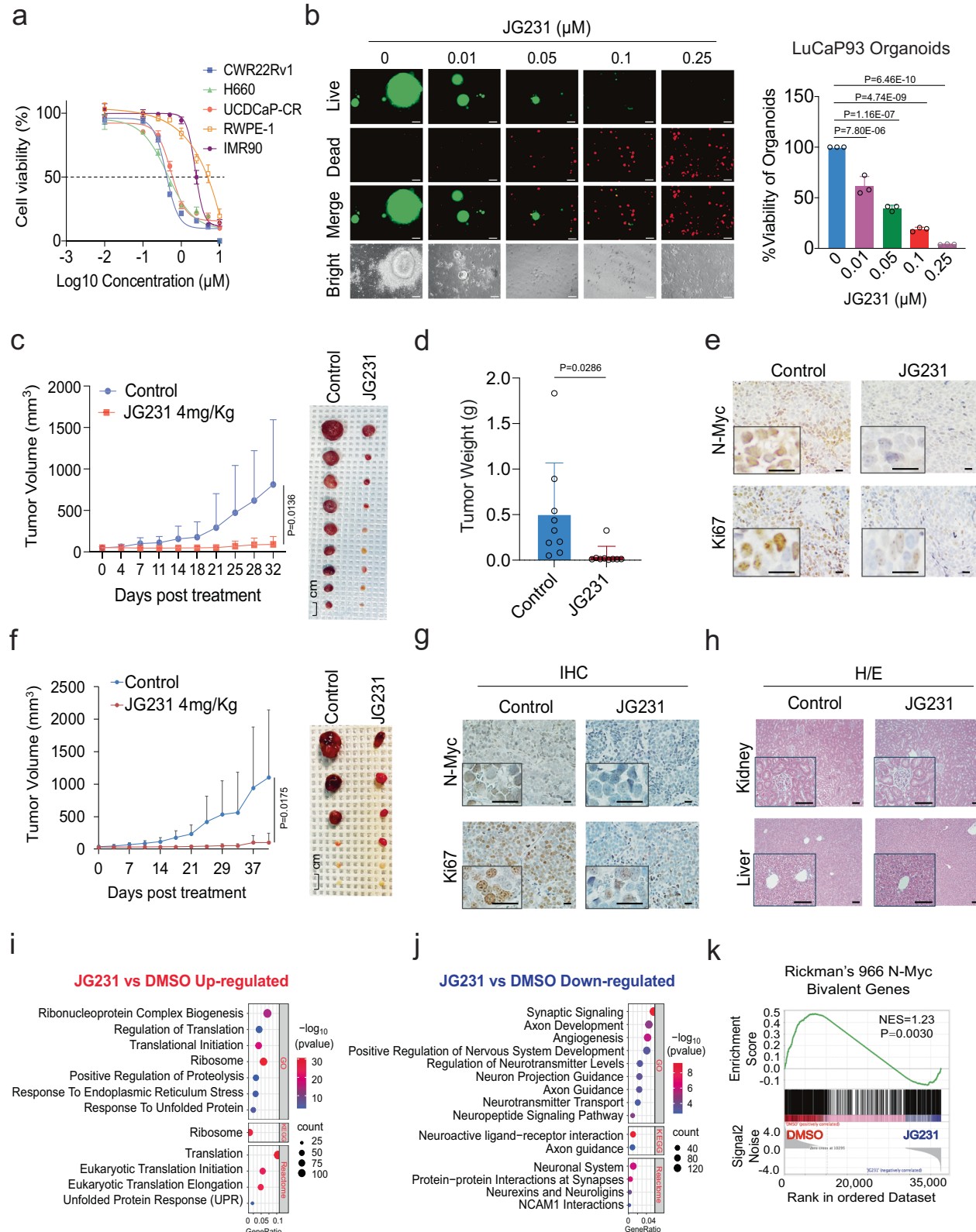

## Methods

### Ethical regulation statement

All the research complies with all relevant ethical regulations. All human sample collection has been complied with all relevant ethical regulations for work with human participants at UC Davis. The Institutional Review Board (IRB) approved protocol (protocol number is GU-001) covered the patient specimen acquisition. All the patient provided permission to access residual tissue through the consent process. All animals used in this study received humane care in compliance with applicable regulations, policies, and guidelines relating to animals. The housing conditions were maintained at ambient temperatures of 22 °C with 55% humidity on average and a 12/12 h light/dark cycle and access to food and water. All experimental procedures using animals were approved by the Institutional Animal Care and Use

**Fig. 6 | HSP70 inhibitor JG231 suppresses NEPC xenograft tumor growth.**
**a** CWR22Rv1, H660, UCDCaP-CR, RWPE-1, and IMR90 cells were treated with increasing doses (0.001, 0.01, 0.1, 0.25, 0.5, 1, 2.5, 5, and 10 μM) of JG231 for 3 days, and the viable cells were counted. Results were compared to controls to generate cell viability. **b** Organoids from LuCaP93 PDX were treated with JG231 (0.01, 0.05, 0.1, and 0.25 μM) for 7 days. Cell viability was assayed by the CellTiter-Glo Luminescent assay and the live and dead cells were visualized by immunofluorescence (n = 3 independent samples). Statistical significance was determined by one-way ANOVA with a Turkey multiple-comparison test. Scale bar represents 100 microns. **c, d** Mice bearing LuCaP93 xenografts were treated with vehicle control, or JG231 (4 mg/Kg i.p) for 30 days (n = 9). Tumor volumes were measured twice weekly (**c**). Tumors were photographed and weighed (**d**). Data represented means ± S.D. from 9 tumors per group. Statistical significance was determined by unpaired two-sided

t-test on day 32. **e** IHC staining of N-Myc and Ki67 in each group was performed. Scale bar represents 20 microns. **f** Mice bearing H660 xenografts were treated with vehicle control, or JG231 (4 mg/Kg i.p) for 40 days (n = 6). Statistical significance was determined by unpaired two-sided t-test on day 40. **g** IHC staining of N-Myc and Ki67 in each group was performed. Scale bar represents 20 microns. **h** H/E staining of kidneys and livers in each group was performed. Scale bar represents 100 microns. **i, j** Gene Ontology (GO), Kyoto Encyclopedia of Genes and Genomes (KEGG), and Reactome analyses illustrate the up-regulated and down-regulated pathways in H660 cells treated with 5 μM JG231 for 24 h. **k** Gene Set Enrichment Analysis (GSEA) showing the enrichment of Rickman's 966 N-Myc Bivalent genes in H660 cells treated with JG231. Results are the mean of three independent experiments (± S.D.). Source data are provided as a Source Data file.

Committee of UC Davis complied with ARRIVE guidelines and ethical regulations and humane endpoints (animal protocol numbers are #19796 and 23256).

## Patient-derived xenografts (PDXs) and conditional reprogramed cell cultures (CRCs)

PDX models UCD1172, UCD1173, and UCD1178 were established from specimens from advanced prostate cancer patients at UC Davis Medical Center. 5-week-old male NOD.Cg-*Prkdc^scid^ Il2rg^tm1Wjl^*/SzJ (NSG, Envigo) mice was inoculated with tumor specimens from patients to establish PDX. Once the xenografts were established, tumors were propagated in 5-week-old male NSG or C.B-17/lcrHsd-*Prkdc^scid^Lyst^bg-J^* (SCID, Envigo) mice to further generate CRCs. Primary cells from malignant human prostate tissue or PDX tumors were isolated. Briefly, tissue was minced and digested with collagenase/hyaluronidase/dispase at 37 °C for 1–3 h. The dissociated cell suspension was filtered through a 100 μm cell strainer and collected. Cells were plated in a mixture of complete F-medium/conditioned medium from irradiated J2 cultures supplemented with 10 μM Y-27632 (MCE, #HY-10071/CS-0131). Subculturing was performed with trypsin treatment when required. Prostate cancer PDX models LuCaP35CR, LuCaP49, LuCaP93, LuCaP145.2, and LuCaP173.1 were acquired from Eva Corey[63,64].

## Cell lines and organoid cultures

LNCaP (American Type Culture Collection, ATCC, CRL-2505), C4-2B (a kind gift from Dr. Leland Chung), C4-2B MDVR (C4-2B enzalutamide resistant), PC3 (ATCC, CRL-1435), DU145 (ATCC, HTB-81), and CWR22Rv1 (ATCC, CRL-2505) cells were maintained in RPMI1640, whereas VCaP, UCDCaP (derived from an 84-year-old male Gleason 10 patient), UCDCaP-CR (UCDCaP castration resistant) and HEK293 (ATCC, CRL-1573.3) cells were cultured in DMEM supplemented with 10% fetal bovine serum (FBS), 100 units/ml penicillin, and 0.1 mg/ml streptomycin. All cell line experiments were performed within 6 months of receipt from the ATCC or resuscitation after cryopreservation. C4-2B MDVR cells were maintained in medium containing 20 μM enzalutamide[65,66]. H660 (ATCC, CRL-5813) cells were grown in RPMI-1640 medium contains 5% FBS, 1% penicillin-streptomycin, 1x Insulin, Transferrin, Selenium Solution (ITS-G, Gibco, #41400045), 10 nM hydrocortisone, 10 nM beta-estradiol, and 1x GlutaMAX (Gibco, #35050-061). For generation of N-Myc-overexpressing C4-2B cell lines, the pcDNA3-HA-N-Myc plasmids were transfected into C4-2B cells for 2 days. Multiple monoclonal cells were screened with G418 (300 μg/ml) for 4 weeks, then the expression of N-Myc was examined by western blotting and RT-PCR, and N-Myc-overexpressing clones were selected and cultured in medium containing 300 μg G418 for the further experiments. All cell line experiments were performed within 6 months of receipt from the ATCC or resuscitation after cryopreservation. All cell lines were routinely tested as mycoplasma-free by PCR and authenticated using the short tandem repeat (STR) method. All cells were maintained at 37 °C in a humidified incubator with 5% carbon dioxide.

For organoid cultures, PDX tumor tissues were collected and cut into 2–4 mm³. Tumors were digested using collagenase IV (Gibco, #17104019) and incubated at 37 °C for 30 min until tumor cells were dispersed. Advanced DMEM (ADMEM) medium supplemented with 1x GlutaMAX, 1 M HEPES, 100 u/ml penicillin, and 0.1 mg/ml streptomycin was added to the cell suspension and then filtered through 40 μm cell strainers to obtain a single-cell suspension. The cells were then centrifuged and resuspended in ADMEM complete medium containing GlutaMAX, 100 units/ml penicillin, 0.1 mg/ml streptomycin, B27, N-Acetylcysteine, Human Recombinant EGF, Recombinant FGF-10, A-83-01, SB202190, Nicotinamide, dihydrotestosterone, PGE2, Noggin, and R-spondin (prepared from 293T-HA-Rspo1-Fc cells). Tumor cells were seeded in a 96-well plate with Matrigel diluted in a 1:3 ratio of ADMEM complete medium and incubated at 37 °C for 30 min to solidify the atrigel complex. Next, ADMEM complete medium mixed with JG231, with or without alisertib (Selleck, #S1133), or MK-8745 (Selleck, #S7065), or AURKA inhibitor (Selleck, #S1451), or tozasertib (Selleck, #S1048), was added to each well. The viability of the organoids was analyzed using the CellTiter-Glo Luminescent assay (Promega, #7570) and visualized by immunofluorescence using the LIVE/DEAD® Viability/Cytotoxicity Assay Kit (Thermo Fisher, #L3224) according to the manufacturer's protocol.

## Compounds

JG231 was synthesized and tested for purity (>95%) by High-performance liquid chromatography (HPLC) and identity by Liquid chromatography-mass spectrometry (LC-MS) and Nuclear Magnetic Resonance (NMR). Alisertib (Catalog# S1133), MK-8745 (Catalog# S7065), AURKA inhibitor (Catalog# S1451), tozasertib (Catalog# S1048), and MG132 (Catalog# S2619) were purchased from Selleck. Cycloheximide (Catalog# 01810) were purchased from Sigma-Aldrich.

## Cell growth assay

CWR22Rv1, H660, and UCDCaP-CR cells were seeded in 12-well plates at a density of $0.2 \times 10^5$ cells/well and cultured overnight. After treatment with different concentrations of JG231 or alisertib, the cells were collected and counted to calculate the percentage of cell viability. CWR22Rv1 and H660 and UCDCaP-CR cells were treated with JG231, alisertib, or a combination of both for 7–10 days, and total cell numbers were counted. H660 cells were treated with JG231, MK-8745, AURKA inhibitor, tozasertib, or a combination of JG231 and one of the other compounds for 10 days, and total cell numbers were counted. CWR22Rv1 and H660 and UCDCaP-CR cells were transfected with siRNA targeting N-Myc, HSP70, AURKA, or a negative control 7 days, and then cells were counted to calculate the cell survival ratio compared with the control group. Synergy distribution map was constructed using the Combenefit software[67].

## Clonogenic assay

CWR22Rv1 cells were seeded in 6-well plates at a density of 500 cells/well and treated with different doses of JG231 and alisertib or MK-8745,

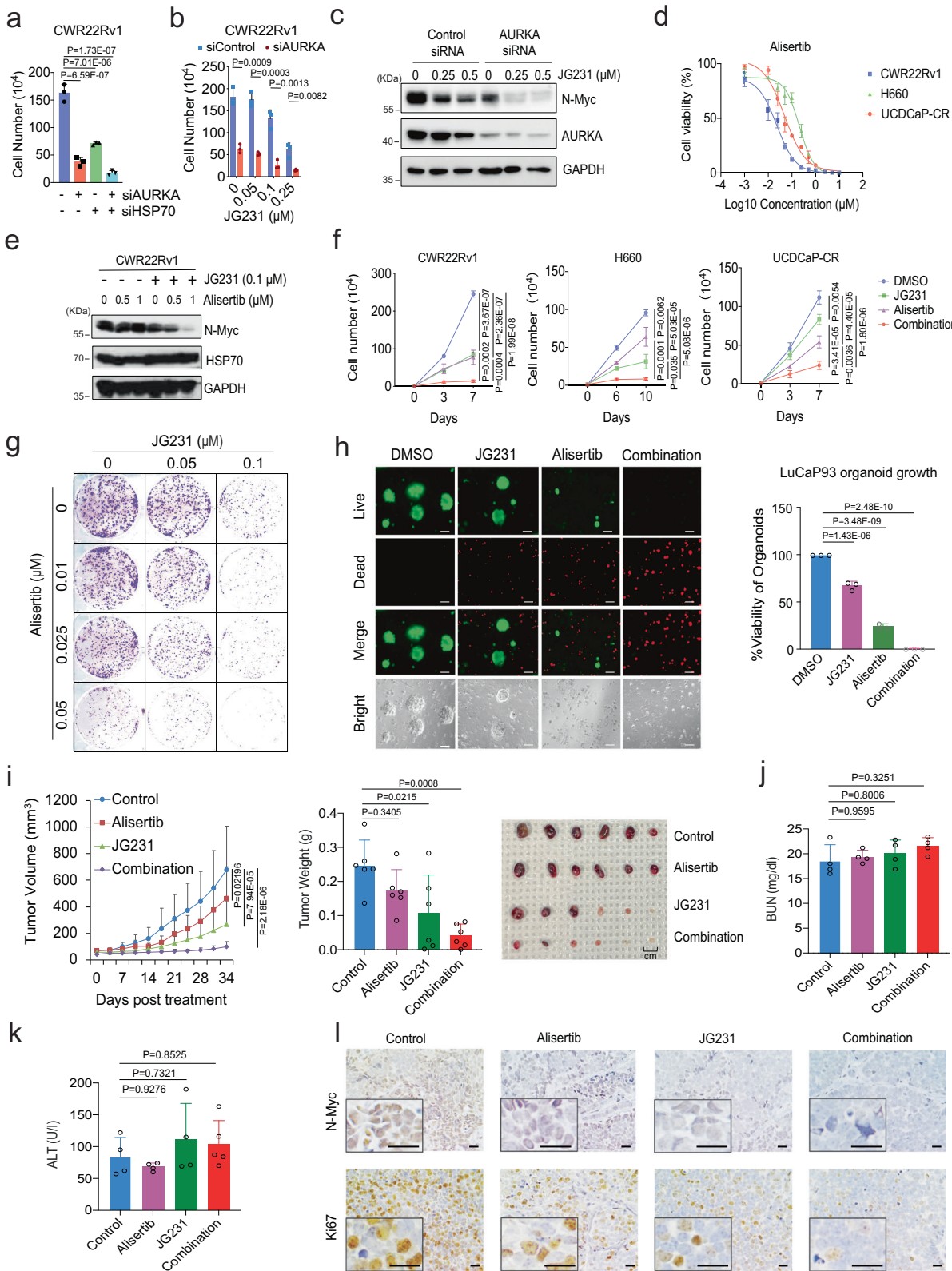

or AURKA inhibitor, or tozasertib for 10 days. After fixing with 4% paraformaldehyde and staining with 0.5% crystal violet for 30 min respectively, the colonies were counted after washing with PBS.

**siRNA, plasmids, and cell transfection**
For small interfering RNA (siRNA) transfection, $2 \times 10^5$ cells were seeded in 60 mm dishes overnight, and then transfected with 20 nM

siRNA targeting the N-Myc, AURKA, HSP70, HSC70, STUB1 sequence (Supplementary Data 3) or control siRNA (Invitrogen, Catalog# 12935300) using Lipofectamine-iMAX (Invitrogen, #13778075). The effect of siRNA-mediated gene silencing was examined using qRT-PCR and western blotting 2–3 days after transfection. Cells were transiently transfected with plasmids expressing HA-N-Myc (Addgene, Catalog#74163), His-N-Myc (SinoBiological, Catalog#

**Fig. 7 | Dual targeting of HSP70 and AURKA enhances N-Myc degradation and improves treatment in NEPC. a** Cell proliferation was measured in CWR22Rv1 cells transfected with siRNA against negative control, HSP70, AURKA, or in combination for 5 days (n = 3 samples). **b, c** CWR22Rv1 cells were transfected with siRNA targeting AURKA or negative control and then treated with JG231. Cell proliferation (**b**) and N-Myc expression (**c**) were determined by cell counting or western blotting, respectively (n = 3 samples). **d** CWR22Rv1, H660, and UCDCaP-CR cells were treated with increasing doses of alisertib for 3 days, and the viable cells were counted. **e** The expression of N-Myc was detected in CWR22Rv1 cells treated with alisertib alone or combined with JG231. **f** CWR22Rv1, H660, and UCDCaP-CR cells were treated with alisertib (0.025 μM for CWR22Rv1, 0.1 μM for other cell lines), JG231 (0.1 μM) or the combination for 7-10 days, and the viable cells were counted (n = 3 samples). **g** CWR22Rv1 cells were treated with JG231 alone or with alisertib in the clonogenic assay. **h** Organoids from LuCaP93 PDX were treated with JG231

(0.1 μM), alisertib (0.25 μM) alone or in combination for 7 days. Cell viability was assayed by the CellTiter-Glo Luminescent assay and the live-and-dead cells were visualized by immunofluorescence (n = 3 samples). Scale bar represents 100 microns. **i** Mice bearing H660 xenografts were treated with vehicle control, JG231 (2 mg/Kg i.p), alisertib (10 mg/Kg p.o), or JG231 plus alisertib for 30 days (n = 6). Tumor volumes were measured twice weekly. Tumors were photographed and weighed. **j, k** Drug toxicity tests were conducted by measuring the blood urea nitrogen (BUN) and alanine aminotransferase (ALT) levels in the serum samples collected from animals in each group (n = 4 samples). **l** IHC staining of N-Myc and Ki67 in each group was performed. Scale bar represents 20 microns. For (**a, f**, and **h–k**), statistical significance was determined by one-way ANOVA with a Tukey multiple-comparison test. For (**b**), statistical significance was determined by an unpaired two-sided t-test. Results are the mean of three independent experiments (± S.D.). Source data are provided as a Source Data file.

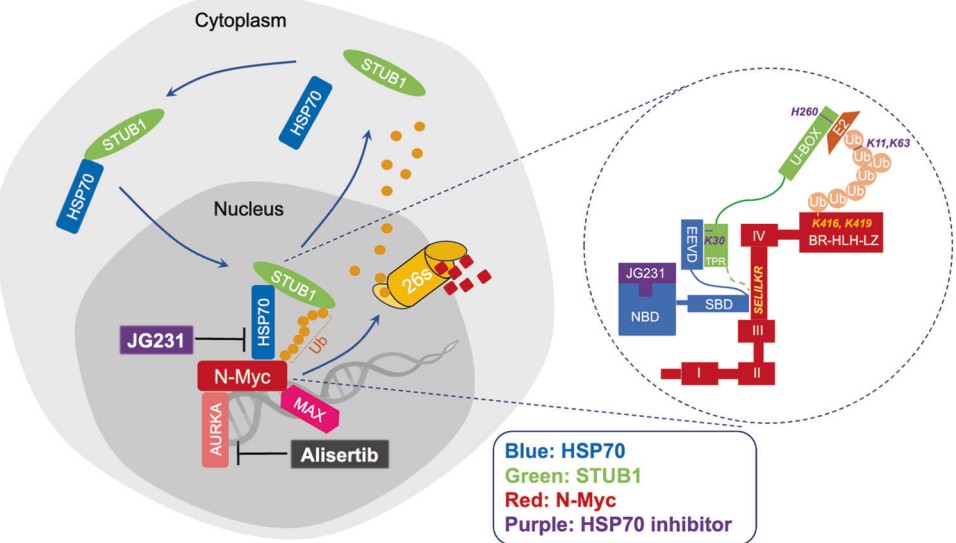

**Fig. 8 | Schematic diagram of the molecular mechanism of HSP70/STUB1 regulating N-Myc proteostasis.** The HSP70/STUB1 complex translocates from the cytoplasm to the nucleus. The substrate binding domain (SBD) of HSP70 occupies the "SELILKR" binding motif on native N-Myc protein, preventing the access of STUB1 and slowing down the turnover of N-Myc protein. However, in the presence of the HSP70 allosteric inhibitor JG231, HSP70 pulls STUB1 closer through binding between the EEVD domain of HSP70 and K30 of the TRP domain of

STUB1. This allows the Ubox of STUB1 to recruit E2 and promote N-Myc ubiquitination at K416 and K419 sites, forming a K11 and K63-linked polyubiquitination chain for degradation in the nucleus and dissociating N-Myc/MAX binding. Dual targeting of AURKA (alisertib) and HSP70 (JG231) causes synergistic degradation of N-Myc and suppresses NEPC tumor growth. Created with BioRender.com, released under a Creative Commons Attribution-NonCommercial-NoDerivs 4.0 International license.

HG17471-CH), Flag-STUB1 (Sino Biological, Catalog# HG12496-NF), HSP70 (OriGene, Catalog# SC116767), Flag-HSP70 (SinoBiological, Catalog# HG11660-NF), His-HSP70 (GenScript, CloneID#OHu15193), Flag-Max (GenScript, CloneID#OHu16927), Flag-AURKA (GenScript, CloneID#OHu23690), pRK5-HA-Ub K11R (Addgene, Catalog#121154), pRK5-HA-Ub K48R (Addgene, Catalog#17604), pRK5-HA-Ub K0 (Addgene, Catalog#17603) or pRK5-HA-Ubiquitin (Addgene, Catalog#17608), using Lipofectamine 2000 (Invitrogen, # 11668019). N-Myc deletion mutant constructs (MYCN Δ1-123, MYCN Δ382-464, MYCN Δ281-464, and MYCN Δ346-464) were generous gifts from Dr. Wei Gu[31].

### Western blot analysis and antibodies

Whole cell protein extracts were resolved on SDS-PAGE, and proteins were transferred to nitrocellulose membranes. After blocking for 1 h at room temperature in 5% milk in PBS/0.1% Tween-20, membranes were incubated overnight at 4 °C with the following primary antibodies: N-Myc (#84406, #51705, 1:1000 dilution for WB, 1:200 for IP, 1:100 for IF, Cell Signaling Technology); c-Myc (#5605, 1:1000, Cell Signaling Technology); L-Myc (#76266, 1:1000, Cell

Signaling Technology); NSE (#MS-335, 1:1000, NeoMarkers); SYP (#PA5-16417, 1:1000, ThermoFisher); CHGA (sc-393941, 1:1000, Santa Cruz Biotechnology); AR (441, sc-7305, 1:1000, Santa Cruz Biotechnology); AURKA (#14475, 1:1000, Cell Signaling Technology); HSP70 (#4873, 1:1000 for WB, 1:200 for IP, Cell Signaling Technology); HSPA8/HSC70 (B-6, sc-7298, 1:1000, Santa Cruz Biotechnology); HSP90 (#ab13492, 1:1000, Abcam), STUB1 (#2080, 1;1000 for WB, 1:100 for IF, Cell Signaling Technology), FLAG® M2 monoclonal antibody (#F1804, 1:1000 for WB, 1:200 for IP, 1:100 for IF, Sigma-Aldrich); His (#66005, 1:5000 for WB, Proteintech); HA (#3724, 1:1000 for WB, 1:200 for IP, 1:800 for IF, Cell Signaling Technology); Ubiquitin (P4D1 and FL76, 1:1000, Santa Cruz Biotechnology); H3 (#ab1220, 1:1000, Abcam), MAX (sc-8011, 1:1000 for WB, 1:100 for IF, Santa Cruz Biotechnology); Tubulin (T5168, 1:5000 for WB, Sigma-Aldrich); GAPDH (#2118, 1:1000, Cell Signaling Technology). Tubulin or GAPDH were used as loading controls. Following incubation with secondary antibodies (W4011 and W4021, 1: 5000, Promega; #5127, 1:2000, Cell Signaling Technology), immunoreactive proteins were visualized using an enhanced chemiluminescence detection system (Millipore, Billerica, MA, USA).

### Real-Time quantitative reverse transcription PCR (RT-qPCR)

Total RNA was isolated using the RNeasy Mini Kit (Qiagen, #74004). cDNA was prepared after digestion with RNase-free RQ1 DNase (Promega, #M6101) and subjected to RT-PCR using SsoFast Eva Green Supermix (Bio-Rad, #1725204) according to the manufacturer's protocol. All reactions were run in Bio-Rad CFX-96 System. Each sample was tested in triplicate and normalized to the co-amplification of actin. The list of primers used in this study is in Supplementary Data 3.

### Co-immunoprecipitation assay

For immunoprecipitation of N-Myc, HSP70, HA, His, or Flag antibodies, an equivalent amount of cell lysate (1500 μg) was immunoprecipitated overnight with 1 μg of the respective antibody and 50 μL Protein A/G agarose (Santa Cruz, #SC-2003) with constant rotation. For His-Tag immunoprecipitation, 50 μl of His-Tag Dynabeads (Invitrogen, #1013D) were incubated with cell lysates for 2 h with rotation according to the instruction. The immunoprecipitants were washed twice with 1 ml 10 mM HEPES (pH 7.9), 1 mM EDTA, 150 mM NaCl, and 1% Nonidet P-40. The precipitated proteins were eluted with 80 μL of SDS-PAGE sample buffer by boiling for 10 min. The eluted proteins were electrophoresed on an SDS-PAGE gel, transferred to nitrocellulose membranes, and probed with the indicated antibodies.

### In vitro ubiquitination assay

In vitro ubiquitination assays were performed according to the manufacturer's protocol. Briefly, we added the 10x E3 Ligase Reaction Buffer (50 mM HEPES, pH 8.0, 50 mM NaCl and 1 mM TCEP), Ubiquitin (100 μM, R&D, #U-100H), UBE1 (100 nM, R&D, #E-305), UBE2D3 (1 μM MCE, #HY-P70998), STUB1 (1 μM, MCE, #HY-P71340), MgATP (10 mM), and His-N-Myc precipitated protein to a 25 μL reaction system and placed in a 37 °C water bath for 3 h. For LC/MS, we terminated the reaction by adding DTT (100 mM) and collected samples for LC/MS analysis. For western blotting, we added protein loading buffer to the samples and boiled them for 10 min before detection.

### Proteomics mass spectrometry analysis using data-independent acquisition (DIA)

Protein concentration was measured by Bradford Plus Protein Assay (Thermo Fisher, #1856210). All protein samples were subjected to clean-up/reduction/alkylation/tryptic proteolysis by using suspension-trap (ProtiFi) devices and resuspended in 50 μL SDS solubilization buffer consisting of 5% SDS, 50 mM TEAB, (pH 7.55). Tryptic digestion constituted of a first addition of trypsin 1:100 enzyme: protein (wt/wt) for 4 h at 37 °C, followed by a boost addition of trypsin using same wt/wt ratios for overnight digestion at 37 °C. To stop the digestion, the reaction mixture was acidified with 1% trifluoroacetic acid (TFA). The eluted tryptic peptides were dried in a vacuum centrifuge and reconstituted in water with 2% acetonitrile (ACN).

For each sample, 500 ng total peptide was loaded onto a disposable Evotip C18 trap column (Evosep Biosytems, Denmark) and subjected to nanoLC on a Evosep One instrument (Evosep Biosystems). Tips were eluted directly onto a PepSep analytical column, dimensions: 150 μm x 25cm C18 column (PepSep, Denmark) with 1.5 μm particle size (100 Å pores) (Bruker Daltronics). Mobile phases A and B were water with 0.1% formic acid (v/v) and 80/20/0.1% ACN/water/formic acid (v/v/vol), respectively. The standard pre-set method of 100 samples-per-day was used, for a 14 min run. The mass Spectrometry was done on a hybrid trapped ion mobility spectrometry-quadrupole time of flight mass spectrometer (timsTOF HT, (Bruker Daltonics, Bremen, Germany), operated in PASEF mode. The acquisition mode was DIA. The acquisition scheme consisted of four 25 m/z precursor windows per 100 ms TIMS scan. Sixteen TIMS scans, creating 64 total windows, layered the doubly and triply charged peptides on the m/z and ion mobility plane. Precursor windows began at 400 m/z and continued to 1100 m/z. Raw files were processed with Spectronaut version 18 (Biognosys, Zurich, Switzerland) using DirectDIA analysis mode. Mass tolerance/accuracy for precursor and fragment identification was set to default settings. The Uniprot database of unreviewed C. Elegans proteins (accessed 28/09/2023 from UniProt, UP0000001940 and a database of 112 common laboratory contaminants (https://www.thegpm.org/crap/) were used. A maximum of two missing cleavages were allowed, the required minimum peptide sequence length was 7 amino acids, and the peptide mass was limited to a maximum of 4600 Da. Carbamidomethylation of cysteine residues was set as a fixed modification, and methionine oxidation and acetylation of protein N termini as variable modifications. A decoy false discovery rate (FDR) at less than 1% for peptide spectrum matches and protein group identifications was used for spectra filtering (Spectronaut default). Decoy database hits, proteins identified as potential contaminants, and proteins identified exclusively by one site modification were excluded from further analysis.

### Dual immunofluorescence assay

$0.5 \times 10^3$ HEK293 cells were plated in 4-well Nunc™ Lab-Tek™ II chamber slides and transfected with HA-N-Myc and Flag-STUB1 for 3 days. After treatment with JG231 and MG132, cells were fixed with 4% paraformaldehyde, permeabilized, and blocked with 0.5% Triton X-100 and 1% bovine serum albumin (BSA), respectively. After slides were washed multiple times with phosphate buffered saline and Tween 20 (PBST), cells were incubated overnight with anti-HA and anti-Flag antibodies. Intracellular HA-N-Myc was visualized using FITC-conjugated secondary antibodies, Flag-STUB1 was visualized using Texas red-conjugated secondary antibodies, and nuclei were visualized with DAPI using an all-in-one fluorescence microscope (BZ-X700).

### Duolink® proximity ligation assay (PLA)

The PLA procedure was performed according to the manufacturer's instructions. Briefly, approximately 500 cells were cultured on 8-well Nunc™ Lab-Tek™ II chamber slides for 24 h and then transfected with HA-N-Myc, Flag-MAX, Flag-STUB1, Flag-HSP70, or His-HSP70 plasmids with lipofectamine 2000 reagent for 48 h and treated with JG231 overnight. Cells were fixed and permeabilized in 4% paraformaldehyde and 0.1% Triton X-100 in PBS for 30 min respectively, followed by blocking with the manufacturer's blocking agent for 1 h and then incubated with paired mixture antibodies of anti-HA and anti-Flag overnight. The next day, after incubation with the corresponding secondary antibody for 1 h at room temperature, staining was performed using Duolink® In Situ Detection Reagents Red (Sigma Aldrich, #DUO92008). His-HSP70 protein levels were assessed using immunofluorescence after the PLA reaction. Fluorescent images were taken on a Zeiss confocal laser microscope system. PLA spots were counted by ImageJ (NIH).

### Site-directed mutagenesis (SDM)

All primers used to generate plasmid constructs encoding for genes with mutations were listed in Table. All mutant constructs were generated using Phusion™ Site-Directed Mutagenesis Kit in accordance with the manufacture's instruction (Thermo Fisher, #F541). Briefly, PCR using a pair of inverse primers with the 5′ end phosphorylated was performed in a 50 μL reaction mixture containing 10 ng plasmids, 0.5 μM forward and reverse primers each, and 0.02 U/μl Phusion Hot Start DNA polymerase. PCR products were digested with DpnI restriction enzyme and ligated with DNA ligase before transformed into competent DH5α E. coli cells and spread onto LB plates with the adequate selective antibiotics. Colonies were picked and screened by DNA sequencing to verify the mutation sites. Primers used for mutagenesis are listed in the Supplementary Data 3. pCMV3-Flag-STUB1

K30A, pCMV3-Flag-STUB1 H260Q, pCMV3-Flag-STUB1 ΔTPR, pCMV3-Flag-STUB1 ΔUBox, pCMV3-HA-MYCN ΔLILKR, pCMV3-HA-MYCN CLPQS, pCMV3-HA-MYCN Δ314-342, pCMV3-HA-MYCN K413R, pCMV3-HA-MYCN K416R, pCMV3-HA-MYCN K419R, pCMV3-HA-MYCN K424R, pCMV3-HA-MYCN K425R, pCMV3-HA-MYCN K444R, pCMV3-HA-MYCN K446R, pCMV3-HA-MYCN K456R, pCMV3-HA-MYCN K457R, pRK5-HA-Ub K6R, pRK5-HA-Ub K27R, pRK5-HA-Ub K29R, pRK5-HA-Ub K33R, pRK5-HA-Ub K63R, pRK5-HA-Ub K11, pRK5-HA-Ub K63 were generated via site-directed mutagenesis and verified by Sanger sequencing.

## Animal studies and treatment regimens

For single treatment experiments, LuCaP93 or H660 PDX tumor fragments of equal size were implanted subcutaneously into the flank of 5 weeks old male C.B-17/lcrHsd-$Prkdc^{scid}Lyst^{bg-J}$ SCID mice (ENVIGO). Tumor-bearing mice (tumor volume about 50–100 mm$^3$) were randomly divided into two groups (8–10 tumors in each group) according to the experimental design. Treatments are as follows: (1) Vehicle control (15% Cremophor EL, 82.5% PBS, and 2.5% DMSO), intraperitoneal (i.p.), (2) JG231 (4 mg/kg, intraperitoneal injection, every other day). For combination treatment experiment, LuCaP93 PDX tumor fragments of equal size were implanted subcutaneously into the flank of 5 weeks old male C.B-17/lcrHsd-$Prkdc^{scid}Lyst^{bg-J}$ SCID mice (ENVIGO). Tumor-bearing mice (tumor volume about 50-100 mm$^3$) were randomly divided into four groups (6 tumors in each group) according to the experimental design. Treatments are as follows: (1) Vehicle control (15% Cremophor EL, 82.5% PBS and 2.5% DMSO), intraperitoneal (i.p.), (2) JG231 (2 mg/kg, intraperitoneal injection, every other day), (3) alisertib (10 mg/kg, orally daily), (4) JG231 (2 mg/kg, intraperitoneal injection, every other day) plus alisertib (10 mg/kg, orally daily). The tumors were measured using calipers twice a week and tumor volumes were calculated using length × width × width × 0.52. Mouse body weight was monitored once per week. Tumor tissues were harvested and weighed after 5–6 weeks of treatment. Tumor and organ tissues were paraffin embedded and H/E stained. The levels of blood urea nitrogen (BUN), phosphorus, calcium, total protein, albumin, globulin, glucose, cholesterol, alanine transaminase (ALT), alkaline phosphatase (ALP), and total bilirubin in mouse serum was determined by FUJI DRI-CHEM 4000 veterinary chemistry analyzer using the FUJI DRI-CHEM Slide Comprehensive S-Panel (Heska, #6330).

## Immunohistochemistry

Tumors were fixed by formalin and paraffin embedded tissue blocks were dewaxed, rehydrated, and blocked for endogenous peroxidase activity. Antigen retrieving was performed in sodium citrate buffer (0.01 mol per Litter, pH 6.0) in a microwave oven at 1000 W for 3 min and then at 100 W for 20 min. Nonspecific antibody binding was blocked by incubating with 10% fetal bovine serum in PBS for 30 min at room temperature. Slides were then incubated with anti-Ki67 (MA5-14520 at 1:500; Thermo Fisher) or anti-N-Myc (#51705 at 1:200; Cell Signaling Technology) at 4 °C overnight. Slides were then washed and incubated with biotin-conjugated secondary antibodies for 30 min, followed by incubation with avidin DH-biotinylated horseradish peroxidase complex for 30 min (Vectastain ABC Elite Kit, Fisher, #NC9864481). The sections were developed with the diaminobenzidine substrate kit (Fisher, #NC9276270) and counterstained with hematoxylin. Nuclear staining of cells was scored and counted in 5 different vision fields. Images were taken with an Olympus BX51 microscope equipped with DP72 camera.

## Whole exome sequencing (WES) and data analysis

DNA was isolated from snap-frozen patient tumors, xenografts, and cell lines using the DNeasy Blood & Tissue Kit (Qiagen) and subsequently quantified with the 1x dsDNA Broad Range Assay Kit on a Qubit fluorometer (Thermo Fisher Scientific). Genomic DNA samples were sheared into fragments of mean peak size of 150–350 bp using a Covaris S220 focused-ultrasonicator. Sequencing libraries were prepared from 500 ng sheared DNA with the KAPA Hyper Prep Kit (Roche) according to the manufacturer's standard protocol for end repair and A-tailing, adapter ligation, high-fidelity PCR amplification, and clean-ups. Exome enrichment of the DNA library (500 ng) was performed by hybrid capture with the IDT xGen Exome Research Panel (427,692 probes, 39 Mb target region, 19,396 genes; Integrated DNA Technologies). Pooled exome libraries were then multiplex sequenced (paired-end, 150 bp) on an Illumina NovaSeq 6000 sequencing system to achieve >100X coverage (-5.0 Gb output/library).

WES data was analyzed utilizing the Illumina DRAGEN (Dynamic Read Analysis for Genomics) ultra-rapid next generation sequencing data analysis platform (Edico Genome)[68] and applying the DRAGEN Somatic Pipeline. Briefly raw sequence data (FASTQ format) was mapped (ALT-aware enabled)/aligned to the GRCh38/hg38 human reference to generate a position sorted and duplicate-marked BAM file. Variant calling was performed with the DRAGEN Haplotype Caller, which identifies variations from the reference by localized haplotype assembly, Smith–Waterman alignment, and then implements read likelihood calculations using a hidden Markov model (HMM). The results were outputted in standard variant call format (VCF) file output. Variants were hard filtered (e.g., QUAL threshold value, low depth) and further filtered for those having a variant allele frequency >0.2. In addition, for analysis of WES data derived from xenograft tissues, as well as patient tumors and cell lines, Xenome was utilized for human/mouse read classification upstream of the DRAGEN pipeline[69]. Variant annotation was performed with VarSeq software (Golden Helix, Inc.) to apply essential information, including gene ID/description, nucleotide and amino acid sequence alteration, variant type (e.g., missense, nonsense, frameshift, insertion, deletion), functional impact predictions (e.g., dbNSFP), and cancer relevance sourced from databases such as the Catalog of Somatic Mutations in Cancer (COSMIC) and The Cancer Genome Atlas (TCGA). Functional enrichment analysis was performed using the ToppFun application of the ToppGene suite[70].

## RNA-seq and data analysis

RNA was extracted from UCDCaP and UCDCaP-CR cells, or H660 cells treated with 2.5 and 5 µM JG231 for 24 h. RNA-seq libraries from 1 µg of total RNA were prepared using the Illumina Tru-Seq RNA Sample according to the manufacturer's instructions. The mRNA-Seq paired-end library was prepared using Illumina NGS on a HiSeq 4000:2 × 150 cycles/bases (150 bp, PE). Around 30 M of reads/sample were obtained. Data analysis was performed using a Top Hat-Cufflinks pipeline and sequence read mapping/alignment was performed using HISAT. StringTie Data were mapped and quantified for unique genes/transcripts. Gene and transcript expression was quantified as FPKM (Fragments Per Kilobase of transcript per million mapped reads). Principal Component Analysis (PCA) was conducted on the FPKM gene-level data for all genes/transcripts that passed the filter (Filtered on Expression > 1, | log2 ((FKPM1 + 0.1)/(DMSO + 0.1), 2)|> 0.25, and kept 0→values and values → 0) in the Raw Data. The genes commonly regulated by JG231 treatments were clustered with the Hierarchical Clustering algorithm using the R.

## Gene set enrichment analysis (GSEA)

GSEA (SeqGSEA, RRID: SCR_005724) was performed using Java desktop software (http://software.broadinstitute.org/gsea/index.jsp)[71,72]. Genes were ranked according to the shrunken limma log2 fold change, and the GSEA tool was used in the 'pre-ranked' mode with all default parameters. Nervous system-related signaling pathways was used for GSEA analysis.

### Datasets and patient cohorts

The expression levels of N-Myc, AR, KLK3, CHGA, and SYP from PDX or patient tumors were examined from GSE160393, GSE199596, and GSE126078. The Rickman 966 gene set was downloaded from GSE117306. All the data were downloaded from NCBI's Gene Expression Omnibus (GEO) database.

### Statistical analysis and reproducibility

Statistical analyses were performed using Graphpad Prism 9. Raw data were summarized by means, standard deviations (SD), and graphical summaries, and then transformed, if necessary, to achieve normality. Sample size was determined based on the power to detect significant differences ($p < 0.05$). No sample or data point from the analysis was excluded. The experiments and data process were not blinded. Data are presented as mean ± SD from three independent experiments. The results of immunoblotting, immunohistochemistry, immunofluorescence, and PLA are representative of at least three biologically independent experiments showing similar results. Differences between individual groups were analyzed using a two-tailed Student's t-test for single comparisons or one-way analysis of variance (ANOVA), followed by the Tukey's procedure for multiple group comparisons. In the tumor growth experiments, size of the tumor at sacrifice serves as the primary response measure. The tumor growth across groups was analyzed by one-way ANOVA. $P < 0.05$ was considered statistically significant.

### Reporting summary

Further information on research design is available in the Nature Portfolio Reporting Summary linked to this article.

## Data availability

The Whole Exome Sequencing data generated in this study have been deposited in the National Center for Biotechnology Information (NCBI) database under accession code PRJNA1050656. The RNA sequence data have been deposited in the GEO database under accession codes: GSE249917 and GSE249916. Proteomics data have been deposited in the Center for Computational Mass Spectrometry at UC San Diego under accession code MSV000093611 (https://nam12. safelinks.protection.outlook.com/?url=http%3A%2F%2Fmassive.ucsd. edu%2FProteoSAFe%2Fstatus.jsp%3Ftask%3Dfbb023432d854b8c93 209cd5d93c5d2b&data=05%7C02%7Ccffliu%40ucdavis.edu%7C146d0 b4331df4c58bf6e08dbf83ef274%7Ca8046f6466c04f009046c8daf9 2ff62b%7C0%7C0%7C638376721464309074%7CUnknown%7CTWFp bGZsb3d8eyJWIjoiMC4wLjAwMDAiLCJQIjoiV2luMzIiLCJBTiI6Ik1haW wiLCJXVCI6Mn0%3D%7C3000%7C%7C%7C&sdata=709MmQYIRGg% 2BVSxD6%2F%2FpY0CsMyaWokOWyY%2FoiHTBWZA%3D&reserved= 0). The nonsynonymous variants from Whole Exome Sequencing generated in this study have been deposited in the Figshare database (https://doi.org/10.6084/m9.figshare.24570337)[73]. The remaining data are available within the Article, Supplementary Information or Source Data file. Source data are provided with this paper.

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

## Acknowledgements

We thank The Genomics Shared Resource Core at UC Davis Comprehensive Cancer Center which is funded by the UC Davis Comprehensive Cancer Center Support Grant (CCSG) from the National Cancer Institute (NCI P30CA093373). We thank Dr. Junwei Zhao in the Department of Biochemistry and Molecular Medicine at UC Davis for the assistance in our study. We thank Dr. Wei Gu from Columbia University for providing *MYCN* deletion mutant constructs. We thank Dr. Daniel S Peeper from The Netherlands Cancer Institute for providing *STUB1* mutant constructs. The maintenance and characterization of the LuCaP PDX models were supported by the Pacific Northwest Prostate Cancer SPORE (P50CA97186), the Department of Defense Prostate Cancer Biorepository Network (W81XWH-14-2-0183), and National Institutes of Health P01-CA163227. This work was supported in part by grants from National Institutes of Health R37CA249108 (C.L.), R01CA251253 (C.L.), R21CA277171 (C.L.), Department of Defense HT9425-23-1-0144 (C.L.), HT9425-23-1-0325 (C.L.), HT9425-23-1-0324 (M.A.D.), National Institutes of Health R01NS059690 (J.E.G.), and Department of Defense PC180716 (J.E.G.).

## Author contributions

P.X., J.C.Y., and C.L. conceived of the project and designed the experiments. P.X., J.C.Y., B.C., G.G., C.G.T., O.R.J., J.E.G., X.Z., and C.L. developed the methodology. P.X., J.C.Y., B.C., S.N., C.N., Y.S., and G.G. performed the experiments and acquired the data. C.N., L.W., R.S., C.P.E., M.A.D., L.L., Q.W., A.C.G., H.C., B.P., O.T.J., E.C., and J.E.G. provided the technical and material support. C.G.T., P.X., B.C., and S.N. performed bioinformatics analysis. P.X., J.C.Y., B.C., G.G., J.E.G., and C.L. interpreted and analyzed the data. P.X., J.C.Y., and C.L. wrote the manuscript. B.C., C.N., M.A.D., and J.E.G. edited the manuscript. C.L. supervised the study.

## Competing interests

The authors declare no competing interests.
