## [Peer Review file · Nature Communications]

Proteostasis perturbation of N-Myc leveraging HSP70 mediated protein turnover improves treatment of neuroendocrine prostate cancer

Corresponding Author: Professor Chengfei Liu

Version 0:

Reviewer comments:

Reviewer #1

(Remarks to the Author)

In this manuscript (#NCOMMS-23-58531-T), Dr. Xu and colleagues discovered that the oncoprotein N-MYC is subjected to proteostatic regulation via the ubiquitin-proteasome system (UPS). Specifically, they propose that the key molecular chaperone HSP70 binds to N-MYC and prevents the access of E3 ligase STUB1, thereby stabilizing N-MYC proteins. At the molecular level, they found that a conserved "SELILKR" sequence on the N-MYC serve as the binding motif of HSP70 and two lysine residues, K416 and K419, are the ubiquitination sites by STUB1.

In vivo, pharmacological HSP70 inhibition impeded the growth of NEPC showing high N-MYC expression, which can be further enhanced by combinatorial AURKA inhibition.

This manuscript presents extensive molecular studies elucidating the N-MYC-HSP70 interactions and N-MYC ubiquitination. However, several caveats weaken their main conclusions, which need to be carefully addressed. Thus, a major revision is recommended.

Major concerns:

1. The HSP70 chaperone family contains several members, with either constitutive or inducible expression. Based on the MS data (Fig. 2C), both inducible (HSPA1B/HSP72) and constitutive (HSPA8/HSC70) interact with N-MYC. The gene nomenclature HSP70 is very vague and should be clearly defined. Importantly, it is not clear which HSP70 family member is knocked down and the sequences of HSP70 siRNAs are not available in the manuscript. This is pivotal to this manuscript since HSPA8/HSC70 chaperones a large number of cellular proteins and maintains the basal protein folding inside cells. It is crucial to distinguish HSP72 and HSC70, regarding their roles in stabilizing N-MYC. It would be very intriguing if only HSP72, but not HSC70, stabilizes N-MYC.

2. The author suggested that HSP70 overexpression mitigates N-MYC ubiquitination and proteasomal degradation by blocking the association of STUB1 E3 ligase with N-MYC. This conclusion is supported by Fig. 2O-2Q. However, the presented evidence is not very convincing. In Fig. 2O, it is intriguing that overexpressing HSP70 reduced FLAG-STUB1 expression and the FLAG-STUB1 was used for co-IP. And, the IPed STUB1 was unequal, which may underlie the reduced N-MYC co-IPed with STUB1. If there is indeed a reduction in STUB1 expression, either N-MYC should be used for IP or the data need to be normalized by STUB1 levels. Also, in Fig. 2P, overexpressed HSP70 should be stained at the same time. In Fig. 2Q, the STUB1 should also be probed. Overall, these key experiments need to be repeated at least three times and quantitation data should be presented. The most convincing experiment would be in vitro reconstitution experiments, like Fig. 2K, to test if HSP70 could compete with STUB1 for N-MYC binding. Alternatively, the authors could knock down HSP70 and detect N-MYC-STUB1 interactions.

3. While the authors suggest that the HSP70 inhibitor JG231 promotes N-MYC degradation by stalling HSP70 with N-MYC and holding STUB1 in proximity. However, in both Fig. 5B and 5H, JG231 diminished HSP70 expression, a finding incompatible with their proposed model in which JG231 action depends upon HSP70 binding to N-Myc. In fact, the steric hindrance model cannot be excluded. Overall, convincing evidence is required to substantiate the mechanism of action of JG231 in destabilizing N-MYC. In Fig. 5F and 5K, to examine if JG231 promotes N-MYC-HSP70 interactions, both HSP70 and STUB1 should be probed following N-MYC co-IP. It is important to examine how JG231 affects the N-MYC-HSP70 interaction.

4. Also, in Fig. 5F it is a bit surprising that overnight JG231 treatment did not increase global protein polyubiquitination, given

that HSP70 inhibition would be expected to cause global protein misfolding and ubiquitination. In Fig. 5J, the IPed STUB1 levels are unequal. This experiment should be repeated and quantitated.

5. In Fig. 5M, JG231 treatment reduced the expression of both N-MYC and MAX. Thus, the co-IP and PLA between N-MYC and MAX are expected to diminish. Not sure, these data are meaningful.

6. Given that JG231 inhibits both inducible and constitutively expressed HSP70 family members (PMCID: PMC6104643), the therapeutic window would be narrow, a major concern for therapeutic inhibition of HSP70. Thus, it is critical to demonstrate the selectivity of JG231 between primary or immortalized cells and transformed cancer cells. Primary or immortalized prostate epithelial cells should be included for cytotoxicity in Fig. 6A.

7. The authors propose that JG231 blocks the in vivo growth of NEPC mainly by degrading N-MYC proteins. However, generally HSP70 inhibition would be expected to cause proteome-wide misfolding and degradation, far beyond N-MYC. Thus, it remains unclear how much this tumor suppression effect of JG231 is mediated through N-MYC degradation. The authors should at very least express a mutant N-MYC at K416 and K419 residues to rescue the viability of NEPC cells treated with JG231. In fact, studying the systemic impact of HSP70 inhibitors on the cellular proteome is more appropriate to advocate their therapeutic implications.

8. To better evaluate the impact of HSP70 inhibitors on cellular proteome, the global protein polyubiquitination should be probed using both the treated tumors and normal mouse tissues.

9. In Fig 2D, the reason for decreased HSP70 levels in HSP70 IP samples is not clear. Also, the IgG sample in lane 3 seems to have signal for HSP70. The authors need to repeat this experiment and improve the washing steps post IP.

10. The authors mentioned in lines 239-240 that STUB1 overexpression had no effect on N-Myc levels while in Fig 3I, the HA-N-MYC seems to be decreased with Flag-STUB1 overexpression (lanes 3 and 4). There is a contradiction between the data and conclusions drawn, which needs to be addressed.

11. In the extended data 5a, and b, the mRNA levels of HSP70 seem to be nearly two-fold increased upon JG231 treatment in both the CWR22Rv1, and H660 cell lines. Also, the HSP70 protein level is decreased in Fig 5B after 1 M JG231 treatment. The authors haven't explained the reasons for these findings.

Minor concerns:

1. All the staining images lack scale bars.
2. For the IP experiments, IgG heavy or light chains should be probed to indicate equal inputs of Abs, especially important for the IgG controls.
3. All protein stability (pulse-chase) experiments seem to only be performed once, since the quantitation data have no error bars.
4. In Fig. 2D, the molecular weights of HSP70 and STUB1 are labeled wrong.
5. The synergistic effect of JG231 and Alisertib seems weak, more like an additive effect.
6. In Line No 89 the authors need to mention that JG231 is the allosteric HSP70 inhibitor.

Reviewer #2

(Remarks to the Author)

Reviewer #3

(Remarks to the Author)

In this manuscript, Xu et al. study proteostasis perturbation by HSP70 inhibition and its consequences on N-Myc in neuroendocrine prostate cancer (NEPC). It is a challenge in field to identify therapeutic strategies targeting transcriptional activity or levels of N-Myc, a driver of neuroendocrine prostate cancer. Here, the authors approach this question through efforts to identify molecular chaperones that might be involved in N-Myc proteostasis. By label-free mass spectrometry they identified HSP70 as one of the highest binding chaperone proteins of N-Myc, found that N-Myc protein turnover is mediated by the HSP70/STUB1 complex, and showed preliminary evidence of this being a potential strategy for the treatment of NEPC.

The authors first develop and characterize a PDX-derived castration resistant cell line model with neuroendocrine features

along the well-known lines in the literature, and confirmed N-Myc to be regulated by UPS in these cells. With an IP pull-down for N-Myc, binding proteins including several HSP70 family proteins were identified.

The authors assess also STUB1, which is not indicated to be altered based on protein identification data shown in Figure 2C. The authors should explain how they hypothesized that STUB1 may be relevant and participating in the HSP70-related regulation and interaction with N-Myc. This is currently not justified (paragraph on lines 174-183).

The work proceeds to show that STUB1 mediates N-Myc protein ubiquitination, to identify interaction sites, and to show that inhibition of HSP70 facilitated the STUB1-dependent ubiquitination and turnover of N-Myc. Then, *in vivo* studies are used to show inhibitory effect of JG231, an allosteric inhibitor of HSP70's ATPase activity, in limiting NEPC tumor growth and improving the efficacy of aurora kinase A inhibitor, alisertib, previously shown to inhibit growth of NEPC tumors by disrupting the N-Myc signaling pathway.

The manuscript presents a substantial body of work of very high quality. The assessments are detailed, thorough, and convincing. The paper is fluently written and easy to read. Methods are described with sufficient detail and the ethics are in place.

Of data availability, it is stated that the data obtained in this study are available upon reasonable request from the corresponding author. This needs to be updated to indicate that proteomics and RNA sequencing data, for which original data should be deposited in public domains according to FAIR principles, are opened to public access once the paper is published. In addition, normalized, filtered gene and protein-wise data used in the analyses need to be provided as supplementary tables in the publication.

The authors have recently published a paper showing relevance of the same HSP70/STUB1 pathway and inhibitors in AR-regulation in AR-positive prostate cancer cells (Xu et al. *Pharmacol Res.* 2023 Mar; 189: 106692). Since this paper argues potential of targeting the identified pathway in treatment-induced NEPC, the manuscript should discuss these in comparison.

Minor points:

In the introduction line 72, there is an unusual phrasing: "In tumors, elevated proteostasis activity..." As proteostasis is a state, not something containing activity, this should be rephrased.

Some of the figure labellings could be more self-explanatory for easier read. E.g. PLA-interaction figures like Fig.2B and 3g should say "interaction" in the figure heading, not just the protein names. Figure 5 label "positive" is also not particularly informative.

Reviewer #4

(Remarks to the Author)

The authors of the manuscript "Proteostasis perturbation of N-Myc by HSP70 inhibition improves treatment in neuroendocrine prostate cancer" report here on a functionally important interaction between the proteins N-MYC, HSP70 and the E3 ubiquitin ligase STUB1 and the role of these proteins in neuroendocrine prostate cancer (NEPC). They first describe the establishment of a new cellular NEPC model by repeated passage of primary prostate carcinoma cells in castrated mice. The resulting cells show high levels of N-MYC and resemble NEPC cells in some respects. These cells are dependent on N-MYC expression and N-MYC is an unstable protein and is rapidly degraded by the UPS. In interactome studies, an interaction of N-MYC with the heat shock protein HSP70 is observed and the rest of the manuscript focusses on the functional relevance of this interaction. After apparently validating the interaction, the authors show that genetic depletion of HSP70 reduces N-MYC levels and that this reduction becomes proteasome-dependent. They then describe that the protective effect of HSP70 on N-MYC levels depends on the E3 ligase STUB1, a known HSP70 interactor. It is described that STUB1 can be an N-MYC E3 ligase and interacts with N-MYC via a specific motif and which lysine residues on N-MYC are necessary for its ubiquitylation. It is then shown that the allosteric HSP70 inhibitor JG231 enhances the interaction between N-MYC and STUB1 and destabilises N-MYC. Finally, it is shown that this inhibitor limits the growth of NEPC cells *in vitro* and in a xenografted setting and behaves synergistically with the Aurora inhibitor Alisertib. This is a very extensive study, but unfortunately it has logical and technical shortcomings. Here are some points.

1: The observations in Figure 1 that N-MYC is mutationally exclusive with other MYC isoforms, is highly expressed in NEPC, N-MYC levels are regulated at the protein level, N-MYC is a very unstable protein and that the driver mutations originally found in the patient are retained in the newly established model are largely trivial.

2: A major claim of this paper is the interaction between N-MYC and HSP70. I am not convinced that this is a robust and functionally important interaction. Especially the Co-IP experiments are not convincing. Figure 2d and following: The crucial control was not shown: N-MYC and non-HSP70-transfected cells -> HSP70 IP, N-MYC Western. Why are the HSP70 IP levels lower in the N-MYC condition? 2e: No convincing interaction! Missing control (no-HSP70 transfection). 3b, f: The "no-STUB1" control is also missing here. 3c: Again, the co-IP is not convincing. 3g: Positive control of the delta-LILKR mutant (e.g. interaction with MAX) is missing. Figure 3J: Blot for His-HSP70 but no mention of HIS-HSP 70 above blot or in legend. Although the N-MYC IP hardly enriches N-MYC, there is as much or even more ubiquitin signal as in the input. How can this be? Empty vector control is missing and blot is overexposed.

3: A key point for me is how much of the JG231 effect shown is N-Myc dependent. In any case, the proteasome after JG231 incubation and siHSP70 treatment must first be compared to untreated cells by mass spectrometry. How many other proteins are down-regulated here like N-Myc, 1,10, 100 or 1000? Can the effect of JG231 be rescued by N-MYC overexpression or

the expression of a STUB1 resistant mutant?

Author Rebuttal letter:

Reviewer #1

In this manuscript (#NCOMMS-23-58531-T), Dr. Xu and colleagues discovered that the oncoprotein N-MYC is subjected to proteostatic regulation via the ubiquitin-proteasome system (UPS). Specifically, they propose that the key molecular chaperone HSP70 binds to N-MYC and prevents the access of E3 ligase STUB1, thereby stabilizing N-MYC proteins. At the molecular level, they found that a conserved SELILKR sequence on the N-MYC serve as the binding motif of HSP70 and two lysine residues, K416 and K419, are the ubiquitination sites by STUB1. In vivo, pharmacological HSP70 inhibition impeded the growth of NEPC showing high N-MYC expression, which can be further enhanced by combinatorial AURKA inhibition.

This manuscript presents extensive molecular studies elucidating the N-MYC-HSP70 interactions and N-MYC ubiquitination. However, several caveats weaken their main conclusions, which need to be carefully addressed. Thus, a major revision is recommended.

Major concerns:

1. The HSP70 chaperone family contains several members, with either constitutive or inducible expression. Based on the MS data (Figure 2C), both inducible (HSPA1B/HSP72) and constitutive (HSPA8/HSC70) interact with N-MYC. The gene nomenclature HSP70 is very vague and should be clearly defined. Importantly, it is not clear which HSP70 family member is knocked down and the sequences of HSP70 siRNAs are not available in the manuscript. This is pivotal to this manuscript since HSPA8/HSC70 chaperones a large number of cellular proteins and maintains the basal protein folding inside cells. It is crucial to distinguish HSP72 and HSC70, regarding their roles in stabilizing N-MYC. It would be very intriguing if only HSP72, but not HSC70, stabilizes N-MYC.

Answer: We acknowledge the reviewer's perspective and have addressed this concern with additional experimental data. To investigate the impact of HSPA8/HSC70 on N-Myc expression, we transfected two HSPA8-specific siRNAs into CWR22Rv1 cells and assessed N-Myc expression using western blotting. As shown in Supplementary Figure 2f left, knockdown of HSPA8 did not affect N-Myc protein expression. Intriguingly, we observed that HSC70 (HSPA8) knockdown led to an increase in HSP70 (HSPA1A/HSPA1B) protein expression. Subsequently, we conducted a combination knockdown of HSC70 and HSP70, revealing that only HSP70 knockdown resulted in decreased N-Myc expression. Moreover, the combination knockdown showed a similar effect on N-Myc expression as HSP70 knockdown alone (Supplementary Figure 2f right). Although HSC70 binds to N-Myc, we hypothesize that its function is primarily related to N-Myc folding and maturation. In contrast, HSP70 appears to play a more prominent role in N-Myc turnover and stabilization. We have incorporated these additional findings into the revised manuscript with additional discussion (page 17).

2. The author suggested that HSP70 overexpression mitigates N-MYC ubiquitination and proteasomal degradation by blocking the association of STUB1 E3 ligase with N-MYC. This conclusion is supported by Figure 2O-2Q. However, the presented evidence is not very convincing. In Figure 2O, it is intriguing that overexpressing HSP70 reduced FLAG-STUB1 expression and the FLAG-STUB1 was used for co-IP. And, the IPed STUB1 was unequal, which may underlie the reduced N-MYC co-IPed with STUB1. If there is indeed a reduction in STUB1 expression, either N-MYC should be used for IP or the data need to be normalized by STUB1 levels. Also, in Figure 2P, overexpressed HSP70 should be stained at the same time. In Figure 2Q, the STUB1 should also be probed. Overall, these key experiments need to be repeated at least three times and quantitation data should be presented. The most convincing experiment would be in vitro reconstitution experiments, like Figure 2K, to test if HSP70 could compete with STUB1 for N-MYC binding. Alternatively, the authors could knock down HSP70 and detect N-MYC-STUB1 interactions.

Answer: We appreciate the comments. Here are the responses to each point. We have redone Figure 2o, and the STUB1 was equally pulled down in the updated data. We have redone the PLA staining in Figure 2p with the HSP70 staining. We have repeated and included the STUB1 band in Figure 2q. Additionally, in response to the reviewer's concern regarding the HSP70/STUB1 interaction with N-Myc, we conducted knockdown experiments targeting HSP70, STUB1, or both in CWR22Rv1 cells. Our findings revealed that HSP70 knockdown notably reduced N-Myc protein expression. However, this effect was significantly attenuated when STUB1 knockdown was applied concurrently, indicating that the reduction in N-Myc induced by HSP70 knockdown is primarily mediated by STUB1 (Supplementary Figure 2k). Moreover, we performed a Co-IP experiment to investigate the N-Myc/STUB1 interaction following HSP70 knockdown. As demonstrated in Supplementary Figure 2l, knockdown of HSP70 significantly enhanced N-Myc ubiquitination and the binding between N-Myc and STUB1.

3. While the authors suggest that the HSP70 inhibitor JG231 promotes N-MYC degradation by stalling HSP70 with N-MYC and holding STUB1 in proximity. However, in both Figure 5B and 5H, JG231 diminished HSP70 expression, a finding incompatible with their proposed model in which JG231 action depends upon HSP70 binding to N-Myc. In fact, the steric hindrance model cannot be excluded. Overall, convincing evidence is required to substantiate the mechanism of action of JG231 in destabilizing N-MYC. In Figure 5F and 5K, to examine if JG231 promotes N-MYC-HSP70 interactions, both HSP70 and STUB1 should be probed following N-MYC co-IP. It is important to examine how JG231 affects the N-MYC-HSP70 interaction.

Answer: We appreciate the insightful perspective provided by the reviewer and concur that the steric hindrance model likely regulates N-Myc proteostasis through the HSP70/STUB1 complex when HSP70 is absent. Our supplementary knockdown data (Supplementary Figure 2k) corroborates that HSP70 knockdown necessitates the recruitment of STUB1 for N-Myc degradation. Additionally, we observed a significant increase in STUB1 and N-Myc binding following HSP70 knockdown (Supplementary Figure 2l), indicating potential competition between HSP70 and STUB1 for binding to the 'SLILKR' motif on N-Myc. Furthermore, we noted that higher doses of JG231 treatment for longer durations resulted in decreased HSP70 protein expression (especially on nuclear protein level) without affecting mRNA expression. This reduction in HSP70 levels may be attributed to STUB1-mediated self-ubiquitination and subsequent degradation, as demonstrated in previous study (PMCID: PMC4112096). However, this phenomenon was observed only with longer or higher-dose JG231 treatments. Our additional data further revealed that endogenous N-Myc protein expression was reduced at lower doses of JG231 treatment without impacting HSP70 protein expression. Notably, only at higher doses of JG231 (1 μ M) was there a slight decrease in HSP70 expression (Figure 5a-b). We also replicated the experiments for Figure 5f and Figure 5j under MG132 conditions, consistently demonstrating that allosteric inhibition of HSP70 by JG231 treatment facilitated the binding of STUB1 and HSP70 to N-Myc and increased its ubiquitination.

4. Also, in Figure 5F it is a bit surprising that overnight JG231 treatment did not increase global protein polyubiquitination, given that HSP70 inhibition would be expected to cause global protein misfolding and ubiquitination. In Figure 5J, the IPed STUB1 levels are unequal. This experiment should be repeated and quantitated.

Answer: While we acknowledge the reviewer's concern regarding the potential global protein polyubiquitination induced by JG231 treatment, we would like to point out that the ubiquitin-proteasome system and autophagy system play vital roles in maintaining cellular ubiquitination status during treatment. To address this concern, we conducted experiments under MG132 conditions, where we blocked the proteasome system. Remarkably, we found that JG231 treatment significantly increased the total polyubiquitination level in the presence of MG132. Importantly, under these conditions, JG231 treatment prolonged the interaction between N-Myc and STUB1, leading to an enhanced binding of HSP70 and STUB1 to N-Myc (Figure 5f). We repeated the experiment depicted in original Figure 5j (Supplementary Figure 5e in revised version), and now observed that STUB1 was equally pulled down.

5. In Figure 5M, JG231 treatment reduced the expression of both N-MYC and MAX. Thus, the co-IP and PLA between N-MYC and MAX are expected to diminish. Not sure, these data are meaningful.

Answer: The binding between Max and N-Myc is indicative of downstream signaling transduction of N-Myc activity. Our co-IP and PLA data confirmed that JG231 not only degraded N-Myc expression but also reduced N-Myc/MAX binding, thereby blocking N-Myc transcriptional activity. Recognizing the straightforward nature of these findings, we have relocated this data to the supplemental section (Supplementary Figure 5i and Supplementary Figure 5j).

6. Given that JG231 inhibits both inducible and constitutively expressed HSP70 family members (PMCID: PMC6104643), the therapeutic window would be narrow, a major concern for therapeutic inhibition of HSP70. Thus, it is critical to demonstrate the selectivity of JG231 between primary or immortalized cells and transformed cancer cells. Primary or immortalized prostate epithelial cells should be included for cytotoxicity in Figure 6A.

Answer: We appreciate the suggestion. We have updated Figure 6a to include the cell viability data of normal fibroblast cell line IMR90 and immortalized prostate epithelial cell line RWPE-1. Notably, both IMR90 and RWPE-1 cells exhibit lower sensitivity to JG231 treatment compared to cancer cells, with IC50 values approximately 5- to 10-fold higher. Furthermore, our findings demonstrate that JG231 treatment has no significant impact on body weight or gross organ morphology in mice, as depicted in Supplementary Figure 6b and 6c. Moreover, histopathological analysis reveals no notable changes in liver morphology or signs of renal inflammation after JG231 treatment, as shown in Figure 6h. Additionally, in the combination treatment with alisertib, toxicity panel data indicate that neither single treatment with JG231 nor the combination treatment affects kidney and liver function (Figure 7j-k and Supplementary Figure 7i). These comprehensive findings collectively support the safety and tolerability of HSP70 inhibition by JG231 in both in vitro and in vivo settings.

7. The authors propose that JG231 blocks the in vivo growth of NEPC mainly by degrading N-MYC proteins. However, generally HSP70 inhibition would be expected to cause proteome-wide misfolding and degradation, far beyond N-MYC. Thus, it remains unclear how much this tumor suppression effect of JG231 is mediated through N-MYC degradation. The authors should at very least express a mutant N-MYC at K416 and K419 residues to rescue the viability of NEPC cells treated with JG231. In fact, studying the systemic impact of HSP70 inhibitors on the cellular proteome is more appropriate to advocate their therapeutic implications.

Answer: We acknowledge the reviewer's perspective regarding the proteome-wide degradation effects of JG231 treatment across different cancer cells. However, it's important to note that N-Myc protein plays a crucial role as a driver protein in NEPC cell proliferation. Our data demonstrate that knockdown of N-Myc significantly suppressed the growth of H660 and UCDCaP-CR cells. Specifically, in UCDCaP-CR cells, N-Myc protein was induced by constitutive androgen deprivation therapy and regulated cell proliferation. To further investigate the role of N-Myc and its ubiquitination-dead mutants in rescuing the viability of UCDCaP-CR cells by the JG231 treatment, we transfected vector, WT-N-Myc, K416R-N-Myc, and K419R-N-Myc into UCDCaP-CR cells and treated them with different concentrations of JG231. Our results indicate that WT-N-Myc partially rescues cell viability following JG231 treatment, while the two N-Myc mutants show greater rescue effects compared to WT-N-Myc (Supplementary Figure 5h). Additionally, we found that JG231 is not able to degrade K416R-N-Myc and K419R-N-Myc mutants (Supplementary Figure 5g) and Co-IP assays further confirmed that JG231 led to reduced ubiquitination on these mutants (Figure 5k). These findings underscore the pivotal role of N-Myc in regulating cell proliferation in NEPC cells and suggest that JG231 effects are partially mediated through N-Myc protein regulation.

To understand the impact of JG231 broadly and systemically on the cellular proteome, we conducted proteomic profiling using LC/MS in N-Myc overexpressed cell lines treated with JG231. We performed the whole cell and nuclei proteomic profiling in C4-2B N-Myc overexpression cells. As shown in Supplementary Figure 5k, N-Myc overexpression induced the elevation of NE markers (NSE and SYP), suggested that N-Myc is functional in transforming CRPC to NEPC. Then C4-2B N-Myc cells were treated by DMSO or 5 μ M JG231 for 4 hours, as we confirmed by western blot, JG231 reduced 90% N-Myc expression in 4 hours treatment (Supplementary Figure 5l). Further analysis of the nuclear fraction confirmed that JG231 treatment significantly degraded N-Myc protein expression within a very short time frame (Supplementary Figure. 5m). Whole cell lysates and nuclear lysates were then subjected to LC/MS for DIA profiling. In whole cell lysates, a total of 3808 proteins were detected. Among these, 494 proteins exhibited decreased levels, while 162 proteins showed increased levels following JG231 treatment (Supplementary Figure 5n). However, in the nuclear lysates, a total of 5524 proteins were detected. Among these, 917 proteins exhibited decreased levels, while 416 proteins showed increased levels following JG231 treatment (Figure 5m). Notably, only the nuclear fraction was able to detect N-Myc protein, possibly due to its rapid turnover and stability issues. Among the downregulated proteins in whole cell lysates, significant alterations were observed in protein stabilization, protein folding, protein binding, ribosome, and mitochondrial function (Supplementary Figure 5o-p, Table S2). However, among the downregulated proteins in nuclear lysates, significant alterations were observed in RNA splicing, protein localization, protein binding, RNA binding, ATPase activity, ribosome, cell cycle,

axon guidance, and peptide chain elongation (Figure 5n-o, Table S3).

8. To better evaluate the impact of HSP70 inhibitors on cellular proteome, the global protein polyubiquitination should be probed using both the treated tumors and normal mouse tissues.

Answer: We evaluated the global protein polyubiquitination status in both H660 and LuCaP93 tumors treated with JG231. Remarkably, we found no significant alteration in global protein polyubiquitination following JG231 treatment in either LuCaP93 or H660 tumors (Supplementary Figure 6a). As previously discussed in question #4, this observation suggests the potential involvement of the ubiquitin-proteasome system and the autophagy system in maintaining cellular ubiquitination levels throughout the treatment period.

9. In Fig 2D, the reason for decreased HSP70 levels in HSP70 IP samples is not clear. Also, the IgG sample in lane 3 seems to have signal for HSP70. The authors need to repeat this experiment and improve the washing steps post IP.

Answer: We have repeated and replaced the original Figure 2d and 2e (now Figure 2e and Figure 2f in the revised manuscript), as suggested by Reviewer 4. In this updated figure, we have included lanes for non-N-Myc, non-STUB1, or non-HSP70 samples.

10. The authors mentioned in lines 239-240 that STUB1 overexpression had no effect on N-Myc levels while in Fig 3I, the HA-N-MYC seems to be decreased with Flag-STUB1 overexpression (lanes 3 and 4). There is a contradiction between the data and conclusions drawn, which needs to be addressed.

Answer: We have repeated Figure 3i, and the data indicate that STUB1 has limited effects on the mutant N-Myc (delta-LILKR and CLPQS).

11. In the extended data 5a, and b, the mRNA levels of HSP70 seem to be nearly two-fold increased upon JG231 treatment in both the CWR22Rv1, and H660 cell lines. Also, the HSP70 protein level is decreased in Fig 5B after 1h JG231 treatment. The authors haven't explained the reasons for these findings.

Answer: As discussed in question #3, longer or higher-dose JG231 treatments can slightly decrease total HSP70 protein expression. However, we found that the treatment insignificantly changed HSP70 mRNA expression, possibly due to the protein-mRNA feedback effect. We have re-run the Real-time PCR to further confirm that JG231 treatment did not significantly alter the mRNA expression of HSP70 (Supplementary Figure 5c).

Minor concerns:

1. All the staining images lack scale bars.

Answer: We have added scale bars in all the images.

2. For the IP experiments, IgG heavy or light chains should be probed to indicate equal inputs of Abs, especially important for the IgG controls.

Answer: We demonstrated the IgG heavy and light chains when probing with antibodies from the same species to detect the targets. For instance, in the original blots of Figure 2e and 2f, both the heavy and light chains were evenly showed in each lane.

3. All protein stability (pulse-chase) experiments seem to only be performed once, since the quantitation data have no error bars.

Answer: We have added the error bars in the pulse-chase experiments.

4. In Figure 2D, the molecular weights of HSP70 and STUB1 are labeled wrong.

Answer: We appreciate the correction.

5. The synergistic effect of JG231 and Alisertib seems weak, more like an additive effect.

Answer: We have generated the drug synergy distribution map (Supplementary Figure 7d). The

data indicate that JG231 and alisertib exhibit synergistic effects.

6. In Line No 89 the authors need to mention that JG231 is the allosteric HSP70 inhibitor.

Answer: We have made the corresponding changes in the revised manuscript.

Reviewer #2

Answer: We appreciate the co-review of our manuscript and the valuable suggestions provided.

Reviewer #3

In this manuscript, Xu et al. study proteostasis perturbation by HSP70 inhibition and its consequences on N-Myc in neuroendocrine prostate cancer (NEPC). It is a challenge in field to identify therapeutic strategies targeting transcriptional activity or levels of N-Myc, a driver of neuroendocrine prostate cancer. Here, the authors approach this question through efforts to identify molecular chaperones that might be involved in N-Myc proteostasis. By label-free mass spectrometry they identified HSP70 as one of the highest binding chaperone proteins of N-Myc, found that N-Myc protein turnover is mediated by the HSP70/STUB1 complex, and showed preliminary evidence of this being a potential strategy for the treatment of NEPC.

The authors first develop and characterize a PDX-derived castration resistant cell line model with neuroendocrine features along the well-known lines in the literature, and confirmed N-Myc to be regulated by UPS in these cells. With an IP pull-down for N-Myc, binding proteins including several HSP70 family proteins were identified.

The authors assess also STUB1, which is not indicated to be altered based on protein identification data shown in Figure 2C. The authors should explain how they hypothesized that STUB1 may be relevant and participating in the HSP70-related regulation and interaction with N-Myc. This is currently not justified (paragraph on lines 174-183).

Answer: We value the reviewer's insightful perspective. HSP70 interacts with multiple chaperone proteins, potentially participating in client protein folding and stabilization. The diverse functions of HSP70 are regulated by various factors, including J-domain proteins (such as HSP40), nucleotide exchange factors (such as Bag1), and tetratricopeptide repeat domain proteins (such as STUB1). Previous studies have assessed protein-protein interactions based on binding affinity scores and buried surface areas to evaluate the potential for inhibiting these interactions. The HSP70/STUB1 interaction has been identified as a potential target for therapeutic intervention. In light of this, our study specifically investigated the role of the HSP70/STUB1 complex in regulating N-Myc. We have added the relevant of STUB1 in the revised manuscript (page 6).

The work proceeds to show that STUB1 mediates N-Myc protein ubiquitination, to identify interaction sites, and to show that inhibition of HSP70 facilitated the STUB1-dependent ubiquitination and turnover of N-Myc. Then, in vivo studies are used to show inhibitory effect of JG231, an allosteric inhibitor of HSP70's ATPase activity, in limiting NEPC tumor growth and improving the efficacy of aurora kinase A inhibitor, alisertib, previously shown to inhibit growth of NEPC tumors by disrupting the N-Myc signaling pathway.

The manuscript presents a substantial body of work of very high quality. The assessments are detailed, thorough, and convincing. The paper is fluently written and easy to read. Methods are described with sufficient detail and the ethics are in place.

Of data availability, it is stated that the data obtained in this study are available upon reasonable request from the corresponding author. This needs to be updated to indicate that proteomics and RNA sequencing data, for which original data should be deposited in public domains according to

FAIR principles, are opened to public access once the paper is published. In addition, normalized, filtered gene and protein-wise data used in the analyses need to be provided as supplementary tables in the publication.

Answer: We appreciate the feedback on our work, and we have now included the data access codes in the revised manuscript. The Whole Exome Sequencing data generated in this study have been deposited in the National Center for Biotechnology Information (NCBI) database under accession code PRJNA1050656. The RNA sequence data have been deposited in the GEO database under accession codes: GSE249917 and GSE249916. Proteomics data have been deposited in the Center for Computational Mass Spectrometry at UC San Diego under accession code MSV000093611. The remaining data are available within the Article, Supplementary Information or Source Data file. Source data are provided with this paper.

The authors have recently published a paper showing relevance of the same HSP70/STUB1 pathway and inhibitors in AR-regulation in AR-positive prostate cancer cells (Xu et al. Pharmacol Res. 2023 Mar; 189: 106692). Since this paper argues potential of targeting the identified pathway in treatment-induced NEPC, the manuscript should discuss these in comparison.

Answer: We appreciate the suggestion and have provided additional discussion in the revised manuscript (page 16).

Minor points:

In the introduction line 72, there is an unusual phrasing: "In tumors, elevated proteostasis activity". As proteostasis is a state, not something containing activity, this should be rephrased.
Answer: We have rephrased the sentence (page 3).

Some of the figure labellings could be more self-explanatory for easier read. E.g. PLA-interaction figures like Figure 2B and 3g should say "interaction" in the figure heading, not just the protein names. Figure 5 label "positive" is also not particularly informative.

Answer: We appreciate the suggestions provided and have made the necessary changes accordingly.

Reviewer #4

The authors of the manuscript "Proteostasis perturbation of N-Myc by HSP70 inhibition improves treatment in neuroendocrine prostate cancer" report here on a functionally important interaction between the proteins N-MYC, HSP70 and the E3 ubiquitin ligase STUB1 and the role of these proteins in neuroendocrine prostate cancer (NEPC). They first describe the establishment of a new cellular NEPC model by repeated passage of primary prostate carcinoma cells in castrated mice. The resulting cells show high levels of N-MYC and resemble NEPC cells in some respects. These cells are dependent on N-MYC expression and N-MYC is an unstable protein and is rapidly degraded by the UPS. In interactome studies, an interaction of N-MYC with the heat shock protein HSP70 is observed and the rest of the manuscript focusses on the functional relevance of this interaction. After apparently validating the interaction, the authors show that genetic depletion of HSP70 reduces N-MYC levels and that this reduction becomes proteasome-dependent. They then describe that the protective effect of HSP70 on N-MYC levels depends on the E3 ligase STUB1, a known HSP70 interactor. It is described that STUB1 can be an N-MYC E3 ligase and interacts with N-MYC via a specific motif and which lysine residues on N-MYC are necessary for its ubiquitylation. It is then shown that the allosteric HSP70 inhibitor JG231 enhances the interaction between N-MYC and STUB1 and destabilises N-MYC. Finally, it is shown that this inhibitor limits the growth of NEPC cells in vitro and in a xenografted setting and behaves synergistically with the Aurora inhibitor Alisertib. This is a very extensive study, but unfortunately it has logical and technical shortcomings. Here are some points.

1: The observations in Figure 1 that N-MYC is mutationally exclusive with other MYC isoforms, is highly expressed in NEPC, N-MYC levels are regulated at the protein level, N-MYC is a very unstable protein and that the driver mutations originally found in the patient are retained in the

newly established model are largely trivial.

Answer: We appreciate the reviewer's comment and would like to clarify the significance of the observations presented in Figure 1. The observations presented in our study regarding the development and characterization of the unique and novel UCDCaP cell line, provide significant insights into the conservation of genetic mutations across patient and patient-derived models. Specifically, our data reveal that the UCDCaP cell line model retains a substantial proportion of somatic mutations present in the original patient sample. This is demonstrated by the significant conservation of mutations between the patient, PDX, and UCDCaP cell line. Furthermore, our functional enrichment analysis highlights the overrepresentation of critical pathways, including apoptosis, cell cycle regulation, androgen receptor signaling, and DNA damage response, among others, in the common set of mutated genes across all models. Key genes implicated in prostate cancer, such as AR, ATM, BRCA1, and TP53, exhibit mutations in both the patient sample and the derived models. These findings underscore the relevance and fidelity of the UCDCaP cell line model in preserving the genetic landscape of the original patient tumor. Therefore, while some unique mutations are observed in each model, the conservation of most patient-derived mutations reaffirms the relevance and utility of these models for studying prostate cancer biology and therapeutic responses. We emphasized these points in the revised version (page 4). Additionally, the high expression of N-Myc in UCDCaP-CR cells further underscores its potential as a therapeutic target. The regulation of N-Myc levels at the protein level highlights its dynamic nature and potential susceptibility to targeted interventions. Collectively, these observations provide valuable insights into the biology of NEPC development and inform potential therapeutic avenues.

2: A major claim of this paper is the interaction between N-MYC and HSP70. I am not convinced that this is a robust and functionally important interaction. Especially the Co-IP experiments are not convincing. Figure 2d and following: The crucial control was not shown: N-MYC and non-HSP70-transfected cells -> HSP70 IP, N-MYC Western. Why are the HSP70 IP levels lower in the N-MYC condition? 2e: No convincing interaction! Missing control (no-SHP70 transfection). 3b, f: The σ -STUB1 control is also missing here. 3c: Again, the co-IP is not convincing. 3g: Positive control of the delta-LILKR mutant (e.g. interaction with MAX) is missing. Figure 3J: Blot for His-HSP70 but no mention of HIS-HSP 70 above blot or in legend. Although the N-MYC IP hardly enriches N-MYC, there is as much or even more ubiquitin signal as in the input. How can this be? Empty vector control is missing and blot is overexposed.

Answer: We appreciate the reviewer's perspective regarding the Co-IP controls. However, it's important to note that in the Co-IP experiment, the absence of the IP target in non-transfected cells would indeed result in the absence of bands in those samples. To mitigate non-specific antibody binding, an IgG control was included. However, in response to the reviewer's feedback, we repeated the Co-IP experiments with the inclusion of non-transfected control groups. In Figure 2e, we included lanes with non-HSP70, non-N-Myc, or both-non transfected cells and conducted Flag-HSP70 IP. The data indicate that HSP70 can only be pulled down in the presence of both N-Myc and HSP70, suggesting binding between N-Myc and HSP70. Similarly, Figure 2f demonstrates binding between STUB1 and N-Myc. These data further confirm that the non-transfected control group is unnecessary for the Co-IP experiment. In Figure 3c, to maintain group consistency with Figure 3c, we included the STUB1-only control and repeated the Co-IP experiment, confirming STUB1's binding to the region in N-Myc containing amino acid residues 281-345. Additionally, we repeated the experiment in Figure 3c, consistent with the original findings that HSP70 binds to the N-Myc region of amino acid residues 281-345. In Figure 3f, we included a non-STUB1 or non-HSP70 group in each WT-N-Myc and mutant N-Myc groups. The data consistently showed that both HSP70 and STUB1 bind less to the N-Myc delta-LILKR mutant. As suggested by the reviewer, we performed PLA for MAX with WT-N-Myc and delta-LILKR mutant, which suggested that both WT and mutant N-Myc remain bound to MAX (Supplementary Figure 3d). In Figure 3j, we appreciate the reviewer's correction. We included His-HSP70 to the panel and re-run the N-Myc band with less exposure. N-Myc was evenly pulled down in each sample, with each group including the vector control plasmid. Typically, only 1-5% of the proteins used for IP are loaded as input to ensure that both the input and IP samples can be exposed on the same membrane. It is also normal for the N-Myc enrichment lane to exhibit more ubiquitin than the input lane.

3: A key point for me is how much of the JJ231 effect shown is N-MYC dependent. In any case, the proteasome after JG231 incubation and siHSP70 treatment must first be compared to untreated cells by mass spectrometry. How many other proteins are down-regulated here like N-Myc, 1, 10, 100 or 1000? Can the effect of JG231 be rescued by N-MYC overexpression or the expression of a STUB1 resistant mutant?

Answer: We appreciate the question. N-Myc plays a pivotal role in NEPC development, yet it remains undruggable due to intrinsically disordered functional domains and the absence of distinct binding pockets on the protein. Our research has uncovered the involvement of HSP70/STUB1 in N-Myc proteostasis, presenting a promising therapeutic avenue for NEPC treatment driven by N-

Myc. Regarding the proteome-wide degradation effects of JG231 treatment across cancer cells, we conducted whole cell proteomic profiling in C4-2B N-Myc overexpression cells to evaluate the cellular proteome changes induced by JG231 treatment. As illustrated in Supplementary Figure 5k, N-Myc overexpression led to an elevation of NE markers (NSE and SYP), indicating its functional role in transforming CRPC to NEPC. Subsequently, C4-2B N-Myc cells were treated with either DMSO or 5 μ M JG231 for varying time points. As confirmed by western blot, JG231 reduced N-Myc expression by 90% within 4 hours of treatment (Supplementary Figure 5l). Further analysis of the nuclear fraction confirmed that JG231 treatment significantly degraded N-Myc protein expression within a very short time frame (Supplementary Figure 5m). Whole cell lysates and nuclear lysates were then subjected to LC/MS for DIA profiling. In whole cell lysates, a total of 3808 proteins were detected. Among these, 494 proteins exhibited decreased levels, while 162 proteins showed increased levels following JG231 treatment (Supplementary Figure 5n). However, in the nuclear lysates, a total of 5524 proteins were detected. Among these, 917 proteins exhibited decreased levels, while 416 proteins showed increased levels following JG231 treatment (Figure 5m). Notably, only the nuclear fraction was able to detect N-Myc protein, possibly due to its rapid turnover and stability issues. However, among the whole cell lysate and nuclear proteins, 43 and 38 proteasome proteins were detected, respectively, none of which were affected by JG231 treatment (Source Data Figure 5m and Supplementary Figure 5n).

Additionally, to further explore the role of N-Myc and its ubiquitination-dead mutants in rescuing UCDCaP-CR cell viability, we transfected vector, WT-N-Myc, K416R-N-Myc, and K419R-N-Myc into UCDCaP-CR cells and treated them with different concentrations of JG231. Our findings indicate that while WT-N-Myc partially rescues cell viability following JG231 treatment, the two N-Myc mutants exhibit greater rescue effects compared to WT-N-Myc. Additionally, we found that JG231 is not able to degrade K416R-N-Myc and K419R-N-Myc mutants (Supplementary Figure 5g) and Co-IP assays further confirmed that JG231 led to reduced ubiquitination on these mutants (Figure 5k). These findings underscore the pivotal role of N-Myc in regulating cell proliferation in NEPC cells and suggest that JG231 effects are partially mediated through N-Myc protein regulation.

Version 1:

Reviewer comments:

Reviewer #1

(Remarks to the Author)

In this revision, all my major concerns have been satisfactorily addressed. Thus, the reviewer recommends the manuscript for publication.

Reviewer #2

(Remarks to the Author)

Reviewer #3

(Remarks to the Author)

The authors have satisfactorily addressed my comments and concerns.

Reviewer #4

(Remarks to the Author)

I raised three concerns about the original manuscript. The authors revised the manuscript and responded to my concerns as follows:

- 1.: Novelty and significance of data presented in Figure 1: The authors replied here: "Key genes implicated in prostate cancer, such as AR, ATM, BRCA1, and TP53, exhibit mutations in both the patient sample and the derived models. These findings underscore the relevance and fidelity of the UCDCaP cell line model in preserving the genetic landscape of the original patient tumor". Again, I would like to ask, why this is a major finding reported in a complete main Figure. It is expected, that point mutations of oncogenic drivers remain during culture.
- 2.: N-MYC-HSP70 interaction: The data showing the interaction between N-MYC and HSP70 is now more convincing.
- 3.: N-MYC dependency of JG231-effect on cells: The authors now provide proteomic data on changes in cellular protein abundance upon JG231-treatment: "Whole cell lysates and nuclear lysates were then subjected to LC/MS for DIA profiling. In whole cell lysates, a total of 3808 proteins were detected. Among these, 494 proteins exhibited decreased levels, while 162 proteins showed increased levels following JG231 treatment". This results is in line with my concern, that JG231 has several N-MYC independent effects on cells. No data shown in the manuscript demonstrates, that perturbation of N-MYC by

JG231 causes the cell-toxic effects observed with JG231, since hundreds of other proteins react in the same way. Thus the message of the title of the Paper "Proteostasis perturbation of N-Myc by HSP70 inhibition improves treatment in neuroendocrine prostate cancer" is actually not proven by experimental results.

Author Rebuttal letter:

July 8th, 2024

We are delighted that our manuscript has been principally accepted by Nature Communications. The following responses address the reviewers' comments in a point-by-point manner.

Reviewer #1

In this revision, all my major concerns have been satisfactorily addressed. Thus, the reviewer recommends the manuscript for publication.

Answer: We appreciate the reviewer's effort and time in evaluating our manuscript.

Reviewer #2

Answer: We appreciate the co-review of our manuscript and the valuable suggestions provided.

Reviewer #3

The authors have satisfactorily addressed my comments and concerns.

Answer: We appreciate the reviewer's effort and time in evaluating our manuscript.

Reviewer #4

I raised three concerns about the original manuscript. The authors revised the manuscript and responded to my concerns as follows:

1.: Novelty and significance of data presented in Figure 1: The authors replied here: "Key genes implicated in prostate cancer, such as AR, ATM, BRCA1, and TP53, exhibit mutations in both the patient sample and the derived models. These findings underscore the relevance and fidelity of the UCDCaP cell line model in preserving the genetic landscape of the original patient tumor". Again, I would like to ask, why this is a major finding reported in a complete main Figure. It is expected, that point mutations of oncogenic drivers remain during culture.

Answer: We appreciate the reviewer's comments. The data regarding the relevance and fidelity of the UCDCaP cell line model in preserving the genetic landscape of the original patient tumor are reported in Supplementary Figure 1.

2.: N-MYC-HSP70 interaction: The data showing the interaction between N-MYC and HSP70 is now more convincing.

Answer: We are glad that we have convinced the reviewer.

3.: N-MYC dependency of JG231-effect on cells: The authors now provide proteomic data on changes in cellular protein abundance upon JG231-treatment: "Whole cell lysates and nuclear lysates were then subjected to LC/MS for DIA profiling. In whole cell lysates, a total of 3808 proteins were detected. Among these, 494 proteins exhibited decreased levels, while 162 proteins showed increased levels following JG231 treatment". This result is in line with my concern, that

JG231 has several N-MYC independent effects on cells. No data shown in the manuscript demonstrates, that perturbation of N-MYC by JG231 causes the cell-toxic effects observed with JG231, since hundreds of other proteins react in the same way. Thus the message of the title of the Paper "Proteostasis perturbation of N-Myc by HSP70 inhibition improves treatment in neuroendocrine prostate cancer" is actually not proven by experimental results.

Answer: We sincerely appreciate the reviewer's suggestion. To better reflect scientific findings, we have now changed the manuscript title to "Proteostasis perturbation of N-Myc leveraging HSP70 mediated protein turnover improves treatment of neuroendocrine prostate cancer".
